# Norepinephrine regulates calcium signals and fate of oligodendrocyte precursor cells in the mouse cerebral cortex

Frederic Fiore[1,3], Khaleel Alhalaseh[1,3], Ram R. Dereddi [1,2], Felipe Bodaleo Torres[1], Ilknur Çoban [1,2], Ali Harb[1] & Amit Agarwal [1,2] ✉

Oligodendrocyte precursor cells (OPCs) generate oligodendrocytes, contributing to myelination and myelin repair. OPCs contact axons and respond to neuronal activity, but how the information relayed by the neuronal activity translates into OPC $Ca^{2+}$ signals, which in turn influence their fate, remains unknown. We generated transgenic mice for concomitant monitoring of OPCs $Ca^{2+}$ signals and cell fate using 2-photon microscopy in the somatosensory cortex of awake-behaving mice. $Ca^{2+}$ signals in OPCs mainly occur within processes and confine to $Ca^{2+}$ microdomains. A subpopulation of OPCs enhances $Ca^{2+}$ transients while mice engaged in exploratory locomotion. We found that OPCs responsive to locomotion preferentially differentiate into oligodendrocytes, and locomotion-non-responsive OPCs divide. Norepinephrine mediates locomotion-evoked $Ca^{2+}$ increases in OPCs by activating α1 adrenergic receptors, and chemogenetic activation of OPCs or noradrenergic neurons promotes OPC differentiation. Hence, we uncovered that for fate decisions OPCs integrate $Ca^{2+}$ signals, and norepinephrine is a potent regulator of OPC fate.

Oligodendrocytes (OLs) are myelinating cells in the central nervous system (CNS). They are post-mitotic cells and can survive for the entire lifespan of an organism[1,2]. Unlike neurons and astrocytes, OLs are continually produced throughout adulthood[3] from a pool of oligodendrocyte precursor cells (OPCs) constituting 5-8% of all brain cells. Although myelination is thought to end in young adults, it remains incomplete in most brain areas and continues throughout the adulthood[4,5] For example, about 20–40% of axons are unmyelinated in the corpus callosum[6], and axonal segments of excitatory and inhibitory neurons exhibit discontinuous myelination in the cortex[7,8]. This partial myelination in the CNS opens up a window for the formation of new myelin sheaths on unmyelinated axonal segments. Newly layered myelin extends metabolic support to neurons[9], fine-tunes conduction velocity of individual axons and consequently influences neural circuit function[10,11]. Hence, myelin remodeling is thought to be yet another form of long-term plasticity contributing to adaptive changes in brain function[12].

OLs can remodel myelin by extending and retracting existing sheaths, or by adding entirely new internodes[13,14]. The most common form of myelin remodeling originates from OPCs differentiating into new OLs, which then myelinate previously unmyelinated axonal segments[13]. Before OPCs differentiate into myelinating OLs, they integrate numerous cues in their local environment including neuronal activity and growth factors[15]. Rather than evolving unique signaling mechanisms, OPCs detect and respond to neuronal activity using the same pathways engaged by neurons during synaptic transmission. OPCs form direct synapses with unmyelinated axons[16,17] and express a wide variety of synaptic and extra synaptic neurotransmitter receptors[18–20], thereby allowing them to engage in rapid communication. Axon-OPCs synaptic junctions are rapidly lost as OPCs begin to

[1]The Chica and Heinz Schaller Research Group, Institute for Anatomy and Cell Biology, Heidelberg University, Heidelberg, Germany. [2]Interdisciplinary Center for Neurosciences, Heidelberg University, Heidelberg, Germany. [3]These authors contributed equally: Frederic Fiore, Khaleel Alhalaseh. ✉e-mail: amit.agarwal@uni-heidelberg.de

differentiate into OLs[19,20]. This implies that, within the oligodendrocyte lineage, OPCs are uniquely endowed with the ability to sense and respond rapidly to electrical and chemical signals emanating from axons[21].

Neuromodulators including acetylcholine (ACh), norepinephrine (NE) and serotonin (5-HT) can influence brain-wide neuronal activity[22]. In addition to the classical neurotransmitters such as glutamate and GABA, which act through local synaptic activity, OPCs in the adult brain might integrate extra-synaptic inputs from neuromodulators to guide their behavior. Indeed, several transcriptomics studies indicate that distinct subtypes of oligodendrocyte lineage cells (OLCs) express a wide variety of these neuromodulator receptors[23,24] which can affect their proliferation and differentiation[25,26]. Notably, NE is a key neuro-modulator involved in crucial biological processes including arousal, attention, memory formation, sleep/wake cycles and stress management[27]. NE has also been shown to regulate the proliferation and differentiation of neural progenitors[28,29]. Although NE imparts immense influence on the neuronal function, its effect on OPC fate and myelination is elusive.

Neuromodulators such as NE exert their influence on target cells through the activation of G-protein coupled receptors (GPCRs), and often involve $Ca^{2+}$ as a second messenger. $Ca^{2+}$ is a universal signaling ion, and intracellular changes in $Ca^{2+}$ concentrations appear to modulate the fate of OPCs by regulating proliferation, differentiation and programmed cell death[30]. Recent single-cell RNA-sequencing studies on rodent brains have identified a dozen cell types constituting the oligodendrocyte lineage (e.g., OPCs, premyelinating OLs and OLs)[23], each expressing a distinct repertoire of neurotransmitter and neuro-modulator receptors which grants them a unique ability to interact with the neighboring cells. During development, OPCs can experience varying levels of local $Ca^{2+}$ changes based on their location, level of synaptic and extra-synaptic connectivity and type of neurotransmitter receptor expressed, which in turn can influence their fate. In the larval zebrafish, it has been shown that, during development, the characteristics of $Ca^{2+}$ signal experienced by OPCs can regulate their differentiation into OLs[31] as well as the stability of newly formed myelin internodes[32,33]. Although these studies highlight the significance of $Ca^{2+}$ signaling in regulating developmental OL generation and myelination, little is known about the characteristics of $Ca^{2+}$ activity or the pathways that regulate $Ca^{2+}$ signals in OPCs in the mature mammalian brain.

Like neurons, the source, location and amplitude of $Ca^{2+}$ signals in OPCs might encode information about their developmental stage and fate progression. Hence, to understand how $Ca^{2+}$ signals regulate the fate of OPCs, we performed high-resolution imaging of intracellular $Ca^{2+}$ transients across the entire oligodendrocyte lineage, monitored cell fate and chemogenetically manipulated OPC $Ca^{2+}$ signals and neuronal activity. To simultaneously study the $Ca^{2+}$ signals and map the fate of OPCs, we developed novel transgenic mouse lines expressing two variants of a genetically encoded $Ca^{2+}$ sensor and a red fluorescent reporter in OPCs, and performed chronic 2-photon microscopy on awake behaving mice. We found that $Ca^{2+}$ signals in OPCs mainly occur within processes, and that OPCs exhibit distinct $Ca^{2+}$ signals while they proliferate or differentiate. We also found that NE release from neurons evokes intracellular $Ca^{2+}$ increases in OPCs and promote their differentiation in oligodendrocytes.

## Results

### Generation and characterization of dual-color transgenic mouse lines for simultaneous fate mapping and $Ca^{2+}$ imaging in OPCs

Unlike neurons and astrocytes, OPCs in vivo are difficult to transfect using recombinant AAVs and there is no small-size promoter sequence that can reliably and efficiently drive a transgene expression in OPCs. Hence, to simultaneously map the fate and study changes in the $Ca^{2+}$ signals of OPCs while they proliferate or differentiate into oligoden-drocytes (OLs), we took a transgenic approach and generated two distinct triple-transgenic mouse lines (Fig. 1 and Supplementary Fig. 1). To capture an entire spectrum of $Ca^{2+}$ transients (fast and slow) in the soma as well in the processes of OPCs, we used a mouse line conditionally expressing a cytosolic variant of ultrafast genetically encoded green fluorescent $Ca^{2+}$ sensor GCaMP6f[34] (*TIGRE2-tTA-LSL-GCaMP6f*) (Fig. 1a). To generate the first triple-transgenic mouse line, we cross bred these GCaMP6f mice with *NG2-CreER* BAC transgenic mice[35] expressing tamoxifen inducible Cre recombinase (CreER) in OPCs, and a Cre-reporter mouse line expressing cytosolic red fluor-escent protein tdTomato[36] (*Rosa26-LSL-tdTomato*) (Fig. 1a). At 3–4 weeks of age, we injected the resulting triple transgenic mice (*NG2-CreER;TIGRE2-tTA-LSL-GCaMP6f;Rosa26-LSL-tdTomato;* in short referred as *NG2-GC6f;tdT*) with tamoxifen to induce the expression of GCaMP6f and tdTomato reporters in OPCs (Fig. 1b). 2–3 weeks post-tamoxifen injection, we characterized the efficiency and specificity of GCaMP6f and tdTomato expression by performing immunohisto-chemical analysis in the somatosensory cortex (S1; barrel cortex) of coronal brain sections from *NG2-GC6f;tdT* mice using antibodies against tdTomato, eGFP (GCaMP6f is a GFP derivative) and several oligodendrocyte lineage cells (OLCs) specific markers (Fig. 1b–e). Our histological analysis revealed that majority of recombined cells co-expressed tdTomato and GCaMP6f (Fig. 1b). However, a small fraction of cells exclusively expressed tdTomato and had a pericyte-like mor-phology (Fig. 1b, arrowheads). In the S1 cortex, almost all tdTomato/GCaMP6f double positive cells co-labeled with a pan oligodendrocyte lineage cells marker Olig2 (Fig. 1c). Both tdTomato and GCaMP6f co-expressed at high levels by PDGFRα + OPCs (Fig. 1d) and ASPA+ OLs (Fig. 1e), and revealed both the soma and the processes of OPCs and OLs. A high expression of tdTomato and GCaMP6f is key for tracking the fate transitions of OPCs based on changes in the cell morphology, and for recording $Ca^{2+}$ transients in soma and the fine processes of OLCs, respectively.

Often, cytosolic reporter proteins do not reach fine processes of glial cells[37], myelin segments[38], and axonal compartments of neurons[39]. Hence, to record $Ca^{2+}$ transients in OPC processes, mature oligoden-drocytes and axonal projections, we generated a novel ROSA26 tar-geted knock-in mouse line which can conditionally express a membrane anchored sensitive variant of GCaMP6 (*Rosa26-LSL-mGCaMP6s*) (Supplementary Fig. 1). Then, we cross bred *NG2-CreER* mice with tdTomato and mGCaMP6s reporter mouse lines to generate the second triple-transgenic mouse line (*NG2-CreER;Rosa26-LSL-mGCaMP6s;Rosa26-LSL-tdTomato;* in short referred as *NG2-mGC6s;tdT*). 3-4 weeks old mice were injected with tamoxifen to induce expression of tdTomato and mGCaMP6s in OPCs (Supple-mentary Fig. 1a). Similar to *NG2-GC6f;tdT* mice, a detailed histological analysis in the S1 cortex of *NG2-mGC6s;tdT* mice revealed that after Cre mediated recombination almost all cells double positive for tdTomato and mGCaMP6s were Olig2+ OLCs (Supplementary Fig. 1b–d). The majority of tdTomato + /mGCaMP6s+ cells were PDGFRα + OPCs (Supplementary Fig. 1e) and a small subset of the double+ cells were ASPA+ OLs (Supplementary Fig. 1f). As expected for a membrane anchored protein, mGCaMP6s outlined the soma and revealed the processes of OPCs and OLs in a great detail, but didn't fill the cytosol (as seen for tdTomato) (Fig. 1f, g).

### $Ca^{2+}$ signals in OPCs are restricted to $Ca^{2+}$ microdomains and are regulated by neuronal activity

Very little is known about the characteristics of $Ca^{2+}$ signals and the pathways activating these signals in OPCs in vivo. To study cell mor-phology and $Ca^{2+}$ signals of cortical OPCs in mice freely engaging in explorative behavior, we adapted a high-precision mouse mobility tracking setup called Mobile HomeCage[40] (mHC). mHC permitted head-fixation and at the same time allowed mice to freely explore their cage (Fig. 2a). To characterize the entire spectrum of fast and slow $Ca^{2+}$ signals in the soma and processes of the OPCs, we used *NG2-GC6f;tdT*

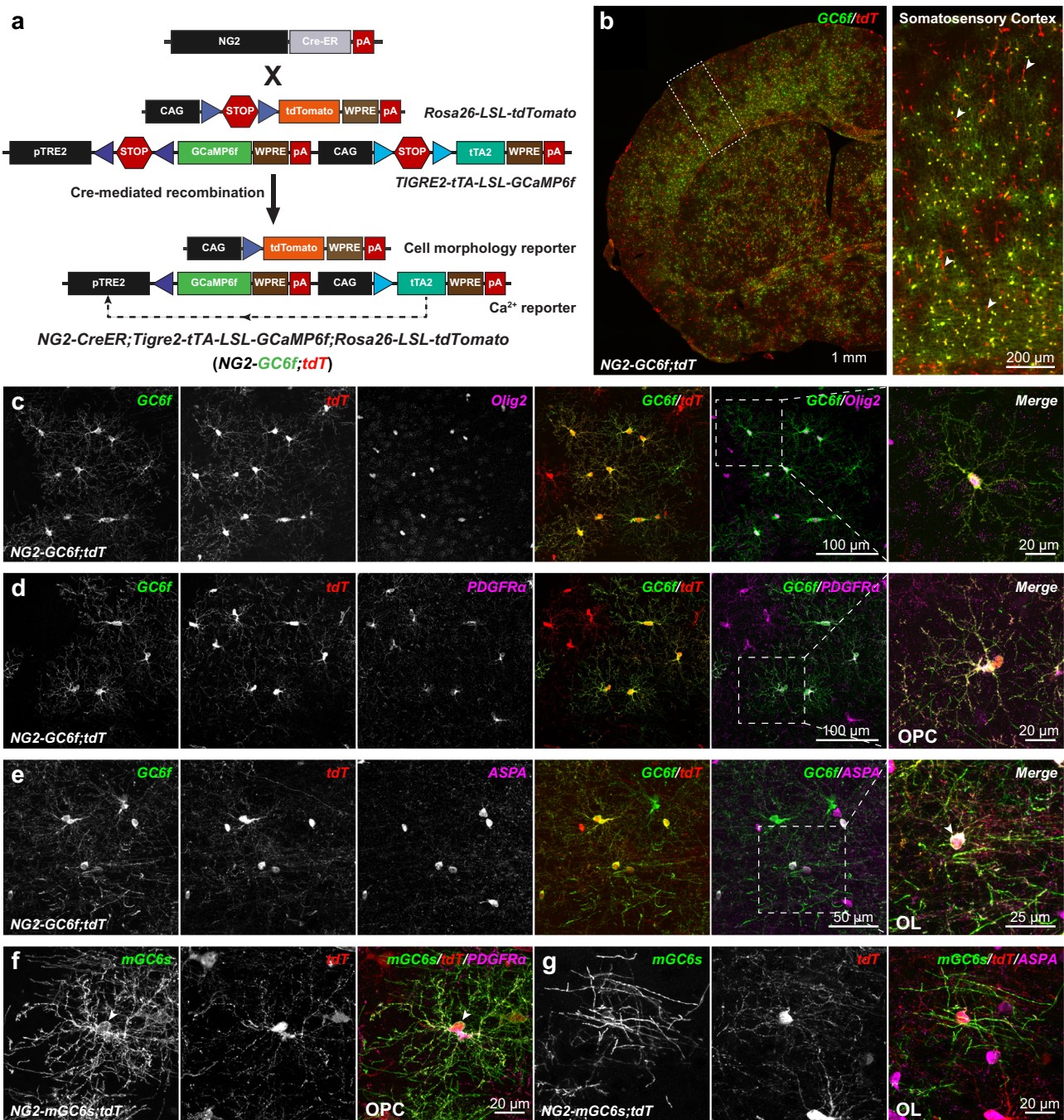

**Fig. 1 | Conditional expression of GCaMP6f, mGCaMP6s and tdTomato in oligodendrocyte lineage cells. a** Cartoon showing a triple transgenic strategy to express the ultrafast variant of GCaMP6 (GCaMP6f) and tdTomato in OPCs using *NG2-CreER;Rosa26-LSL-tdTomato;TIGRE2-tTA-LSL-GCaMP6f* (*NG2-GC6f/tdT*). **b** (left) Coronal brain hemi-section from a *NG2-GC6f/tdT* mouse stained for GCaMP6f (GC6f, green) and tdTomato (tdT, red). (right) Zoom-in image from the somatosensory cortex (boxed area shown in the left panel) shows oligodendrocyte lineage cells co-expressing GC6f and tdT. Arrowheads show recombined pericytes were mostly positive for tdT, but not for GC6f. Scale bars, 1 mm (left) and 200 μm (right). **c** Confocal microscopy images show double recombined (GC6f⁺/tdT⁺) cells express pan oligodendrocyte lineage marker Olig2. (right) High-magnification of the boxed area in **c**. Scale bars, 100 μm (left) and 20 μm (right). **d** Images showing a subset

GC6f⁺/tdT⁺ cells were PDGFRα⁺ OPCs. (right) High-magnification image of GC6f⁺/tdT⁺ OPC in the boxed area in (**d**). Scale bars, 100 μm (left) and 20 μm (right). **e** Images showing a subset of recombined GC6f⁺/tdT⁺ cells expressing ASPA, a marker for mature oligodendrocytes. (right) High-magnification of a GC6f⁺/tdT⁺ oligodendrocyte in the boxed area in (**e**). The arrow highlights the cell body of a recombined mature oligodendrocyte. Scale bars, 50 μm (left) and 25 μm (right). **f** A recombined OPC from the triple transgenic mouse *NG2-CreER;Rosa26-LSL-tdTomato;ROSA-LSL-mGCaMP6s* (*NG2-mGC6s;tdT*) in the somatosensory cortex immunostained for PDGFRα, mGCaMP6s and tdT. The arrow highlights the membrane localization of mGC6s (green) in OPCs. Scale bar, 20 μm. **g** A recombined cell from *NG2-mGC6s;tdT* cortex expressing the mature OL marker ASPA. Scale bar, 20 μm.

mice expressing a cytosolic GCaMP6f, which is capable of capturing Ca²⁺ transients in the millisecond range[41]. After tamoxifen induced expression of tdTomato and GCaMP6f in OPCs, we implanted a glass cranial window over the S1 cortex, and trained mice to remain head-

fixed in the mHC environment (Fig. 2a, details in the "Methods" section). In adult mice (>9 weeks old), we performed a high-resolution dual color 2P-microscopy of OPCs at a 5 Hz image acquisition rate through the cranial window. Since we simultaneously imaged cell

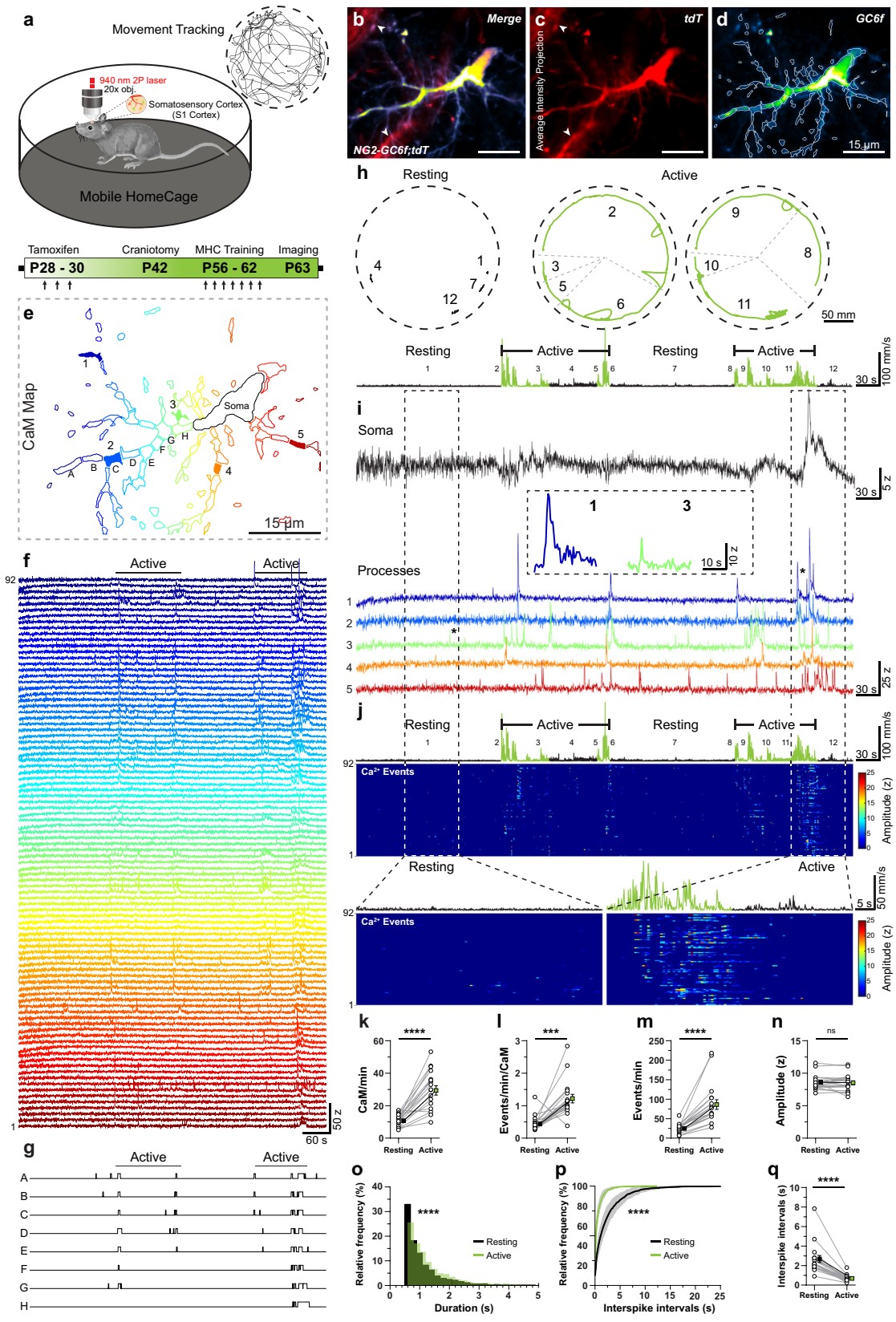

morphology (Fig. 2b, c) and Ca²⁺ signals in a single OPC (Fig. 2b, d), static tdTomato signals enabled a pseudo 'ratiometric' analysis of Ca²⁺ signals and the correction of mouse movement artifacts. We adapted our machine-learning based analysis software CaSCaDe[42] to implement algorithms to automatically correct movement artifacts and analyze Ca²⁺ signals and mouse locomotion activity (see "Methods" for details).

We observed that Ca²⁺ transients in OPCs mostly occur in the processes and remain restricted to small segments – we termed such spatially confined micrometer size (4.64 ± 3.33 μm²) regions in OPCs as Ca²⁺ microdomains (CaMs) (Fig. 2d–f). Binarized Ca²⁺ signals traces from neighboring CaMs indicate that each CaM can exhibit changes in Ca²⁺ concentration independently, but can also occasionally be activated

**Fig. 2 | Characteristics of microdomain Ca²⁺ transients in OPCs in the cortex during baseline and activity state of mice. a** Schematic representation of the Mobile HomeCage (mHC) setup and the experimental timeline. Example 2D tracking data is shown in the top right corner. **b–d** Pseudocolored average intensity projections of an OPC co-expressing cytosolic tdTomato (tdT, **c**) and GCaMP6f (GC6f, **d**) in a *NG2-GC6f;tdT* mouse. The map of active Ca²⁺ microdomains is overlaid on GC6f image **d**. Arrowheads indicate tdT-expressing pericytes (**b**, **c**). **e** Color-coded map of active Ca²⁺ microdomains (CaMs). **f** Color-matched intensity vs time Ca²⁺ traces of the 92 CaMs shown in (**e**). **g** Binary event-detection traces from neighboring CaMs labelled (A–H) in (**e**). **h** 2D tracking data of the location and speed of the mouse in mHC (dashed line) during resting (black: left) or active (green: middle and right) phases. Numbers (1–12) represent distinct resting and activity bouts. **i** Ca²⁺ transients in the soma (top, black) or processes (bottom, see 1-5 color-coded in **e**) of an OPC in vivo. (inset) Examples of detected Ca²⁺ events (asterisks on traces 1 and 3). **j** Heat-map showing the intensity and temporal distribution of Ca²⁺ events aligned with the mouse activity trace (top). Periods of locomotion are highlighted (green, Active). (bottom) Enlarged heat-maps showing Ca²⁺ events during a resting phase (left, bout #1) or an active phase (right; bout #11). Graphs comparing the number of active CaMs per minute (**k**), frequency of Ca²⁺ events per minute per CaM (**l**), frequency of Ca²⁺ events per minute (**m**) and amplitude of Ca²⁺ events (**n**) between the resting and active phases. (**o**) Frequency distribution histogram of the duration of Ca²⁺ events during resting (grey) and active (green) phases. **p** Cumulative frequency distribution of interspike intervals between consecutive Ca²⁺ events during resting (black) and active (green) phases. **q** Graph comparing between the average interspike intervals between the resting and active phases. All data are presented as mean ± SEM. Data from $n = 18$ cells, $N = 3$ mice (**k–n**) Wilcoxon matched-pairs signed rank tests: ***$p = 0.0007$, ****$p < 0.0001$. **o–q** Kolmogorov–Smirnov Test for cumulative frequency distribution: ****$p < 0.0001$. ns, not significant. mm, millimeter; s, seconds; z, z-score. Scale bars, 15 μm. Source data are provided in the Source Data file.

alongside several other CaMs (Fig. 2e: A–H; Fig. 2g), thereby suggesting that CaMs in OPCs can act as independent signaling units.

While we imaged OPC morphology and Ca²⁺ activity, we tracked the location, trajectory, and locomotion speed of the mouse in real time (Fig. 2a, h). The motion tracking data enabled us to precisely correlate microdomain Ca²⁺ activity in OPCs with the spatial location and locomotion of the mouse in mHC (Fig. 2h–j). Over a 10-min recording interval, we observed that somatic Ca²⁺ transients in OPCs were a rare occurrence, and were exclusively seen during intense locomotion activity (Fig. 2i). Next, we found that only a small number of CaMs (10.71 ± 3.55 CaMs/min) are activated in OPCs during the 'resting' bouts (i.e., when mice remain at one location for more than 30 s) (Fig. 2j, bouts 1, 4, 7, 12) with each CaM sparsely firing Ca²⁺ transients (0.44 ± 0.24 events/min/CaM) (Fig. 2j–l). Conversely, when the mouse engaged in explorative bouts of locomotion ('active' phase), the number of activated CaMs in OPCs tripled (29.4 ± 12.07 CaMs/min) (Fig. 2j, k) and each CaM exhibited nearly 3 times more Ca²⁺ transients (1.22 ± 0.56 events/min/CaM) than during the resting phase (Fig. 2j, l and Supplementary Video 1). While the mean frequency of microdomain Ca²⁺ transients increased significantly (resting: 24.89 ± 11.92 events/min and active: 86.16 ± 53.09 events/min), there was no change in the average amplitude of Ca²⁺ events (resting: 8.61 ± 1.17 z and active: 8.52 ± 1.31 z) between the resting and the active phases (Fig. 2m, n). During locomotion however, the average duration of microdomain Ca²⁺ transient increased slightly (resting: 1.27 ± 1.71 s and active: 1.44 ± 1.82 s) (Fig. 2o). In addition, interspike interval (ISI) calculations indicate that, during the active phase, 80% of Ca²⁺ events occurred within 1 s of each other (Fig. 2p). On average, successive Ca²⁺ events occurred at 0.68 ± 0.35 s intervals during the active phase, but were much sparser (2.67 ± 0.68 s intervals) during the resting phase (Fig. 2q).

To further capture the diversity of Ca²⁺ signals generated by OPCs, we imaged Ca²⁺ transients in individual OPCs twice on the same day at acquisition speeds of 5 Hz (Supplementary Data Fig. 2a, b) and 15 Hz (Supplementary Fig. 2a, c) while mice freely engaged in explorative behavior. Indeed, we found that fast microdomain Ca²⁺ transients ranging between 330 and 600 ms can be detected in OPCs, when imaged at a 15 Hz sampling frequency (Supplementary Fig. 2d–g). Such fast Ca²⁺ transients (0.33–0.60 s) constituted about 9.7% and 7.0% of all Ca²⁺ events during resting and active phase, respectively (Supplementary Fig. 2h, i). While imaging at 15 Hz, we saw a significant increase in the number of CaMs and the frequency of Ca²⁺ events while mice actively explored the cage, which is similar to what was observed at 5 Hz (Supplementary Fig. 2j–l). Although the frequency of fast Ca²⁺ events (<0.6 s) increased slightly in response to locomotion, the change in frequency of 'slower' Ca²⁺ events (>0.6 s) accounted for the majority of the increase during locomotion (Supplementary Fig. 2j, l). Taken together, these results suggest that in vivo OPCs exhibit a wide variety of fast and slow Ca²⁺ signals, which are mostly restricted to CaMs in the processes, and that OPC Ca²⁺ transients increase significantly during active explorative behavior.

## Microdomain Ca²⁺ signals correlate with the fate of OPCs

As OPCs divide into daughter cells or differentiate into OLs, they undergo profound changes in their cellular connectivity, neurotransmitter receptor expression, and overall response to neuronal activity[20,43,44]. To study whether OPCs generate distinct Ca²⁺ signals as they proliferate and differentiate, we used *NG2-GC6f;tdT* mice to concurrently track the lineage progression (through changes in the cell morphology) of OPCs and their Ca²⁺ activity at each stage of OPC fate. We took advantage of our *NG2-GC6f;tdT* mouse line, in which tdTomato and GCaMP6f are permanently expressed once OPCs undergo Cre-mediated recombination, even after they divide or differentiate. This implies that we could image cells across the many distinct stages of the OL lineage. Hence, to classify OLCs across developmental stages based on their distinct morphology, we performed immunohistochemistry on the brain sections from adult *NG2-GC6f;tdT* mice and labeled recombined cells with cell-type specific markers for OPCs (PDGFRα+), premyelinating oligodendrocytes (pmOL: BCAS1+; PDGFRα− and ASPA−) and mature OLs (ASPA+). We then acquired confocal z-stacks of the cytosolic tdTomato in each of those cell types (Supplementary Fig. 3a–c). To closely match morphological features of the cells imaged in a single focal plane in vivo, we traced the morphology of OPCs, pmOL and OLs on 2D maximum intensity projected z-stacks (Supplementary Fig. 3d–g). During the analysis, it became evident that three main morphological features (i.e., circularity and area of soma, along with the distribution of primary processes around the soma) can be used in combination to accurately classify cells as OPCs, pmOLs or a mature OLs (Supplementary Fig. 3j–l). To study the properties of Ca²⁺ signals and correlate them with cell-fate, we traced 2D morphology of 167 OLCs we imaged in vivo using the endogenous tdTomato signals (Fig. 3a–d). We classified these cells as OPCs, pmOL and mature OL on the basis of the circularity of the soma (Fig. 3e), its area (Fig. 3f) and the primary process distribution (Fig. 3g). Next, OPCs were further classified based on the ratio between their frequency of Ca²⁺ events (events/min) during active and resting phases (A/R ratio). When the Ca²⁺ events frequency increased to >150% in response to locomotion (i.e. A/R ratio of 1.5), cells were considered as locomotion-responsive (Fig. 3h). Based on this criterion, we could intuitively classify OPCs into two groups: (1) locomotion non-responsive OPC: LNR-OPCs ($n = 29$ cells; mean A/R ratio = 1.13 ± 0.28) (Fig. 3h); and (2) locomotion responsive OPCs: LR-OPCs ($n = 74$ cells; mean A/R ratio = 2.71 ± 1.49) (Fig. 3h). We noted that mainly LNR-OPCs underwent cell division (5 division events out of 29 cells recorded) (Supplementary Fig. 2m, n). When OPCs split into two daughter cells, the total number of CaMs and the frequency of Ca²⁺ events of both the daughter cells together were equivalent to that of the original cell (Supplementary Fig. 2o–q), thereby implying that daughter cells can inherit cellular

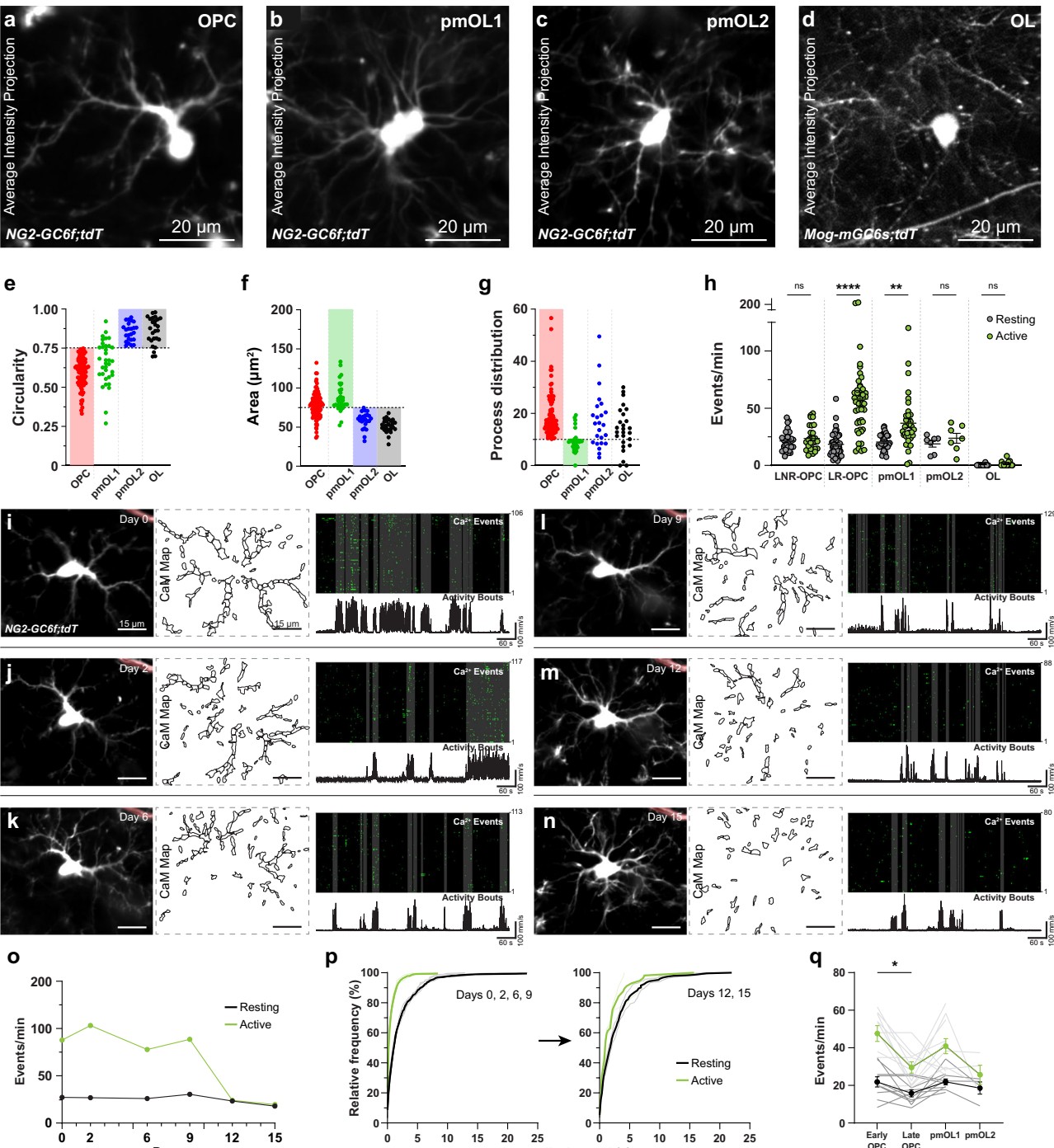

**Fig. 3 | Evolution of Ca²⁺ signals as OPCs progress in their fate.** Average intensity projections of a tdT-expressing OPC (**a**), pmOL1 (**b**), pmOL2 (**c**) and OL (**d**) in vivo. **e**–**g** Scatter plots showing the distribution of soma circularity (**e**), soma area (**f**) and primary process distribution around soma (**g**) of OLCs at distinct developmental stages. Horizontal dotted lines represent threshold values used to segregate cell-types of OLCs. Each cell-type is highlighted with a colored rectangle - OPC, red; pmOL1, green; pmOL2, blue; OL, black. (**h**) Scatter plot showing frequency of Ca²⁺ events at distinct developmental stages during resting (black) and active (green) bouts of activity. **i** (left) Average intensity projection of an OPC (tdT) at the start (Day 0) of the chronic imaging session, and its associated microdomain Ca²⁺ activity (CaM) map. (right) Binarized raster plot showing detected Ca²⁺ events aligned with the locomotion activity trace (bottom). Vertical grey bars in the raster plot represent locomotion bouts. A recombined pericyte (highlighted in red) was used to locate the cell. **j**–**n** Morphological changes, CaM maps, microdomain Ca²⁺ transients and locomotion activity bouts of the OPC shown in **i** at Day 2 (**j**), Day 6 (**k**), Day 9 (**l**), Day 12 (**m**), and Day 15 (**n**). **o** Graph showing changes in the frequency of Ca²⁺ events during the resting and active phases over that period (shown in **i**–**n**; Day 0 – Day 15). **p** Cumulative frequency distributions of interspike intervals during the resting (black) and active (green) phases. Grey lines correspond to individual time points, and thicker black (resting) and green (active) lines represent the average. **q** Graph showing frequency of Ca²⁺ events in tracked OLCs as they undergo differentiation. All data are presented as mean ± SEM. **e**–**h** $n = 169$ cells, $N = 11$ mice. Kruskal–Wallis test: **$p = 0.0068$, ****$p < 0.0001$. **q** $n = 11$ tracked cells, $N = 4$ mice; Mixed-effects ANOVA with Sidak's multiple comparisons test, *$p = 0.0240$. ns not significant. mm, millimeter; s, seconds. Scale bar, 20 μm (**a**–**d**) and 15 μm (**i**–**n**). Source data are provided in the Source Data file.

connectivity and the repertoire of neurotransmitter receptors from their mother cell (Supplementary Video 2). In addition, the mother and daughter cells often don't exhibit increase in Ca²⁺ transients in response to locomotion (Supplementary Fig. 2r).

Based on our morphological criteria, we could sort pmOLs as well into two classes, pmOL1 and pmOL2, with the latter sharing morphological features closer to mature OLs (Fig. 3e–g). pmOL1 had a baseline Ca²⁺ events frequency similar to that of OPCs, and exhibited significant increase in the frequency of Ca²⁺ transients in response to locomotion, albeit at reduced level compared to OPCs (Resting: $20.35 \pm 6.90$ events/min and Active: $37.51 \pm 23.6$ events/min; A/R: $2.07 \pm 0.98$) (Fig. 3h, Supplementary Fig. 4a–f; Supplementary Video 3). In contrast, pmOL2 don't show any increase in the frequency of Ca²⁺ events in response to locomotion (Resting: $18.89 \pm 7.39$ events/min and Active: $23.76 \pm 11.43$ events/min; A/R: $1.23 \pm 0.35$) (Fig. 3h). Although we could track the lineage progression of OLCs through morphological criteria, it was not trivial to ascertain exactly when pmOLs turned into mature myelinating OLs in vivo. Hence, to directly study the properties of Ca²⁺ signals in mature OLs, we generated a new mouse line by crossbreeding transgenic mice expressing Cre under control of mature OL specific promoter Mog (Mogi-Cre)[45] with the conditional mGCaMP6s and tdTomato reporter mice (*Mogi-Cre;Rosa26-LSL-mGCaMP6s; Rosa26-LSL-tdT; Mog-mGC6s;tdT*). We imaged Ca²⁺ transients in mature OLs in the S1 cortex of awake *Mog-mGC6s;tdT* mice aged 8–10 weeks (Supplementary Fig. 4g–l). As expected, the morphological features of the OLs labeled in *Mog-mGC6s;tdT* mice fitted well with the morphological criteria we defined to classify OLs (Fig. 3e–g, black). A detailed Ca²⁺ signal analysis showed that, during a 10-min imaging session, only about 50% of the OLs showed Ca²⁺ transients (Supplementary Fig. 4m). Mature OLs exhibited very few baseline Ca²⁺ transients ($0.65 \pm 0.85$ events/min), and these signals were mostly restricted to the processes and myelin segments, as somatic Ca²⁺ transients were rarely seen (Supplementary Fig. 4j). Ca²⁺ transients in mature OLs were distinct from OPCs, especially with their slower kinetics and prolonged duration (Duration: $2.96 \pm 3.13$ s) (Supplementary Fig. 4k, l). Mature OLs did not exhibit a significant increase in the frequency of Ca²⁺ events in response to locomotion ($2.26 \pm 2.39$ events/min) (Supplementary Fig. 4n).

To systematically track the lineage progression of OPCs, we imaged tdTomato and GCaMP6f signals in individual OPCs in the S1 cortex every 2–4 days over a period of 2–3 weeks (Fig. 3i–n). A detailed analysis of their Ca²⁺ signals showed that OPCs maintained very stable microdomain Ca²⁺ activity (Fig. 3o, Resting) with few morphological reorganizations over several days (Day 0, 2, 6, and 9; Fig. 3i–l). During this stable period, OPCs remained responsive to locomotion-induced increases in the neuronal activity, and increased frequency of events by approximately 3-fold (Fig. 3o, Active). As the OPC differentiated into a pmOL, we noted prominent morphological changes as seen by a circular soma and increased cellular complexity, similar to what we have defined through histological criteria (Fig. 3m, n: Day 12 and 15). In addition, pmOLs (Days 12, 15) did not exhibit any increase in its frequency of Ca²⁺ events (Fig. 3o) in response to locomotion. Also, ISIs between the resting and active phases of exploratory behavior indicate a loss of responsiveness to the locomotion once the OPC (Days 0, 2, 6, 9: Fig. 3p) differentiated into a pmOL (Days 12 and 15: Fig. 3p). The population analysis of 27 OPCs we tracked, suggested that mostly LR-OPCs (i.e., ones highly responsive to locomotion) tend to differentiate. We also noticed a drop in the frequency of locomotion-evoked Ca²⁺ events between the moment when the cells were first imaged (Early OPC; $47.62 \pm 13.22$ events/min) and just as they were ready to differentiate (Late OPC; $29.55 \pm 9.51$ events/min) (Fig. 3q). Furthermore, when cells further progressed towards a more differentiated state such as pmOL2, they stopped responding to locomotion altogether (Fig. 3q). Taken together, these results further reinforce our observations that OPCs exhibit unique Ca²⁺ signatures at different stages

during their lineage progression. LR-OPCs highly responsive to locomotion-evoked neuronal activity tend to differentiate into OLs, and those non-responsive (LNR-OPCs) to this mode of activity likely undergo cell division.

## Activation of glutamate and GABA receptors contribute to the baseline but not the locomotion-evoked Ca²⁺ signals

OPCs express a wide variety of ionotropic and metabotropic receptors for neurotransmitters and neuromodulators including glutamate and GABA[18]. To study the effect of neuronal activity and ionotropic glutamate and GABA receptor activation on baseline and locomotion-induced Ca²⁺ signals in OPCs, we performed in vivo 2-photon Ca²⁺ imaging in the S1 cortex. We first imaged Ca²⁺ signals from a select population of OPCs (18 cells) (Supplementary Fig. 5a, c), then blocked ionotropic glutamate and GABA receptors in vivo by i.p. injecting a mixture of brain accessible antagonists to these receptors (NBQX, 10 mg/kg; CGP39551, 10 mg/kg; Bicuculline; 4 mg/kg). 30–40 min post-injection, we imaged Ca²⁺ signals in the same population of OPCs (Supplementary Fig. 5b, d). We found that an acute block of glutamate and GABA receptors in vivo resulted in about 39% reduction in the frequency of baseline Ca²⁺ signals in OPCs (Supplementary Fig. 5e). However, these antagonists did not block the locomotion-induced increase in Ca²⁺ signals and did not affect their amplitude (Supplementary Fig. 5f–h). These results suggest that local neuronal release of glutamate and GABA contribute to the baseline microdomain Ca²⁺ activity in OPCs, but locomotion-induced Ca²⁺ signals in OPCs are independent of ionotropic glutamate and GABA receptor activation.

To further dissect the source of baseline Ca²⁺ transients in OPC, we performed 2-photon Ca²⁺ imaging in the S1 cortex in acute brain slices derived from 6–12 weeks old *NG2-mG6s;tdT* mice (Supplementary Figs. 1 and 5). Like in vivo, baseline Ca²⁺ transients in OPCs were spatially restricted to discrete microdomains in acute brain slices (see below). However, the number of activated CaMs ($1.89 \pm 1.78$ CaMs/min) and the frequency of Ca²⁺ transients ($3.79 \pm 3.8$ events/min) during spontaneous activity in acute brain slices (Supplementary Fig. 5i, j, k) were about 50% and 15% of the baseline activity (compare: Fig. 2k–m; $10.71 \pm 3.55$ CaMs/min and $24.89 \pm 11.92$ events/min) seen in vivo, respectively. These spontaneous Ca²⁺ transients persisted in OPCs when the neuronal activity was blocked by a voltage-gated sodium channel blocker tetrodotoxin ($0.5 \, \mu M$; TTX). There were almost no changes in the number of CaMs (TTX: $1.92 \pm 1.46$ CaM/min) or the frequency of Ca²⁺ events (TTX: $3.64 \pm 3.65$ events/min), but a slight increase in the amplitude of Ca²⁺ events in the presence of TTX (Supplementary Fig. 5i–l). In addition, about 24% and 30% of the spontaneous Ca²⁺ transients in OPCs were blocked by ionotropic and metabotropic glutamate receptors (Glut-B: TTX, CNQX, D-AP5, MCPG) and GABA receptor (Gaba-B: TTX, Gabazine, CGP55845) antagonists respectively, suggesting spontaneous Ca²⁺ transients in OPCs are in part generated by the spontaneous release of glutamate and GABA (Supplementary Fig. 5m–t). The source of these remaining low frequency spontaneous microdomain Ca²⁺ events ($2.85 \pm 1.15$ events/min), which constitute about 10% of the frequency of the baseline events seen in vivo, remains unknown, and can be possibly driven by activation of other cell-surface receptors and cell-intrinsic mechanisms.

## Norepinephrine release induces microdomain Ca²⁺ transients in OPCs

In rodents, locomotion activates neurons in the locus coeruleus (LC), which in turn release norepinephrine (NE) across several brain regions, including the cortex[46]. Since axonal projections of LC neurons in the cortex are very thin and profusely branched, cytosolic variants of GCaMPs don't reach high enough concentration in axons for a reliable measurement of Ca²⁺ signals in vivo[47]. We therefore used a membrane anchored, sensitive variant of GCaMP6 (mGCaMP6s) to image Ca²⁺ signals in LC axons. We cross-bred *Dbh-Cre* mice[48] with the mGCaMP6s

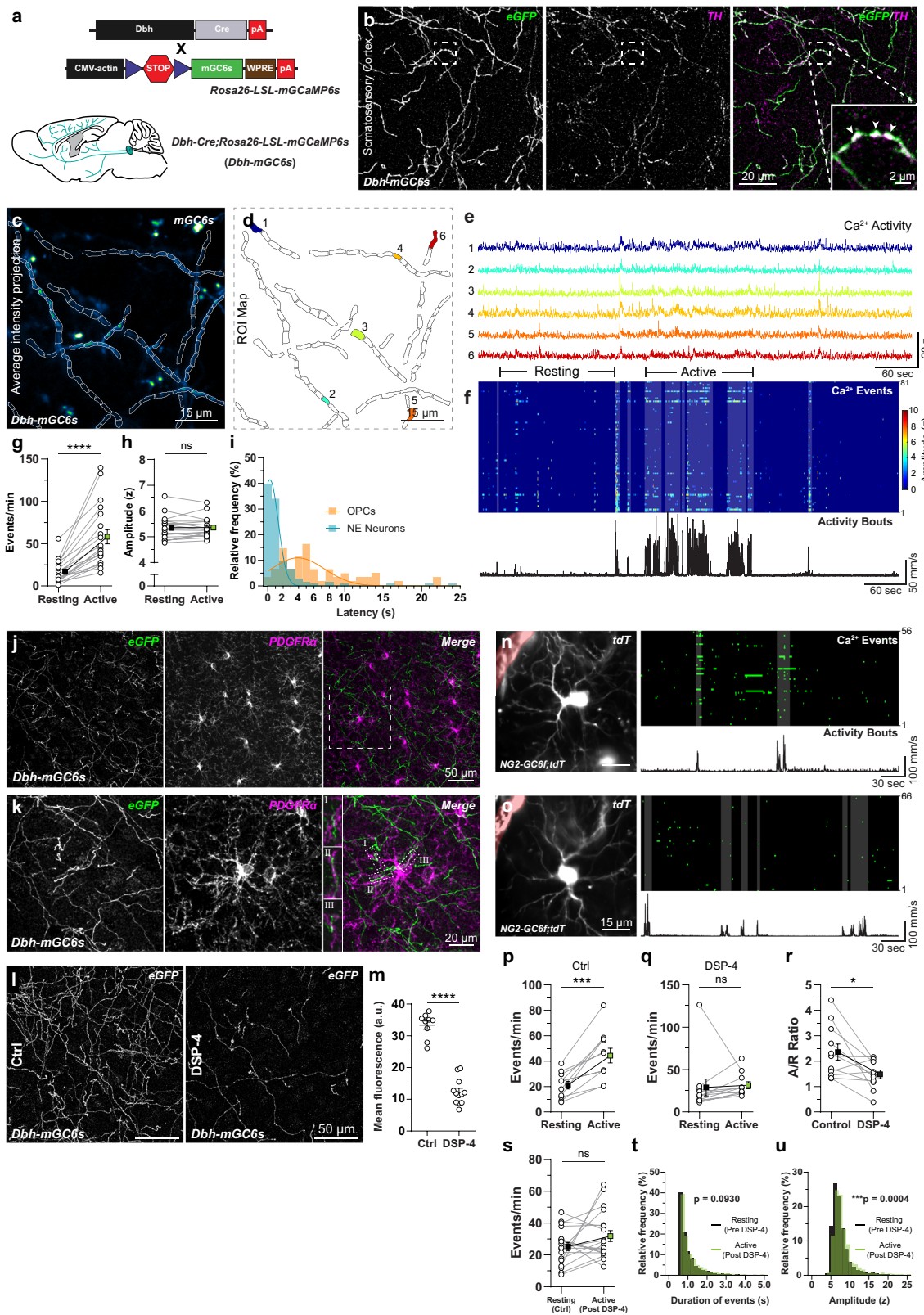

mouse line to generate double transgenic mice (*Dbh-Cre;Rosa26-LSL-mGCaMP6s; Dbh-mGC6s*) (Fig. 4a; Supplementary Fig. 1a). The immunohistochemical analysis in the S1 cortex of adult *Dbh-mGC6s* mice confirmed that all axons expressing mGCaMP6s (labeled with anti-eGFP antibody) were also positive for tyrosine hydroxylase, an enzyme involved in NE synthesis and labeling adrenergic neurons in the cortex (Fig. 4b). Similar to the in vivo Ca²⁺ imaging experiments in OPCs, we

implanted a chronic cranial window on the S1 cortex of 6–8 weeks old *Dbh-mGC6s* mice (time line: Fig. 2a). To assess whether locomotion bouts during explorative behavior in the mHC can engage LC neurons, we imaged Ca²⁺ transients of LC axonal projections in the S1 cortex (Fig. 4c, d). We found that when the mouse was in a 'resting' state, LC neurons fired occasionally (16.72 ± 13.4 events/min) (Fig. 4e–g). However, as soon as mice started to actively explore, as seen by locomotion

**Fig. 4 | Locomotion-induced Ca²⁺ transients in OPCs is mediated by nor-epinephrine. a** (top) Transgenic strategy to express mGCaMP6s in NA neurons in *Dbh-mGC6s* mice. (bottom) Cartoon showing the distribution of NA projections from the LC to the cortex. **b** Maximum intensity projection of mGCaMP6s-expressing (green, eGFP) NA axonal projections (magenta, TH) in the cortex. (inset) Arrowheads highlights varicosities. **c, d** Average intensity projection of NA fibers in the cortex of *Dbh-mGC6s* mice, overlaid with region with Ca²⁺ signals **c**, and color-coded map of detected regions of interest (ROIs) (**d**). **e** Example Ca²⁺ traces of the 6 ROIs shown in (**d**). **f** Heat map of the intensity and temporal distribution of Ca²⁺ events (top), aligned with the mouse locomotion activity (black trace, bottom). Grey highlights show correlation between locomotion and Ca²⁺ signals. Graphs comparing frequency (**g**) and amplitude (**h**) of Ca²⁺ events in NA fibers during resting and active phases. **i** Frequency distribution of the time-lag between the initiation of locomotion and Ca²⁺ activity in NA neurons (cyan) and OPCs (orange). **j** Maximum intensity projection of NA fibers (green) and PDGFRα (magenta) in the cortex. **k** High-magnification image of PDGFRα + OPC and NA fibers in the boxed area in **j**. Insets (I–III) highlight contact between OPC and NA fibers. **l** Maximum intensity projection of NA fibers in control (Ctrl) and DSP-4 treated *Dbh-mGC6* mice. **m** Graph showing the mean fluorescence intensity of eGFP in the cortex of control and DSP-4 mice. **n, o** (left) Average intensity projection of an OPC in a *NG2-GC6f;tdT* mouse before (**n**) and after (**o**) DSP-4 treatment. (right) Raster plots showing Ca²⁺ events (top) and locomotion activity (bottom). A recombined pericyte (highlighted in red) was used to locate the cell (**n, o**). **p, q** Graphs showing the frequency of Ca²⁺ events during the resting & active phases before (**p**) and after (**q**) treatment with DSP-4. **r** Graph showing Active/Resting Ca²⁺ activity ratio (A/R ratio) before and after DSP-4. **s–u** Graph comparing the average frequency of Ca²⁺ event (**r**), the duration (**t**) and the amplitude (**s**) of Ca²⁺ events in the resting phase before DSP-4 (black) and in the active phase after DSP-4 (green). All data are presented as mean ± SEM. **g, h** $n = 19$ imaging fields, $N = 2$ mice; Wilcoxon matched-pairs rank tests: ****$P < 0.0001$. **m** $n = 9$ section, $N = 3$ mice (Ctrl) and $n = 12$ section, $N = 4$ mice (DSP4); Unpaired t-test: ****$P < 0.0001$. **p–s** $n = 11$ cells, $N = 3$ mice; Paired t-test: ***$P = 0.0004$, *$P = 0.0187$ (**p, r, s**), Wilcoxon matched-pairs rank test (**q**). **t, u** $n = 19$ cells, $N = 3$ mice; Kolmogorov–Smirnov Test for cumulative frequency distribution: ***$P < 0.0004$. Ctrl, control. s, seconds. z, z-score. Scale bars: 20 μm and inset 2 μm (**b**); 15 μm (**c, d, n, o**); 20 μm (**k**); 50 μm (**j, l**). Part of the illustration in (**a**) was created using BioRender. Source data are provided in the Source Data file.

bouts (Fig. 4f), the firing rate of LC neurons increased by 3.5 times (57.91 ± 36.9 events/min), but there was no change in the amplitude of Ca²⁺ events between the resting and active phases (Fig. 4g, h; Supplementary Video 4). Additionally, cortical projections of LC neurons mostly fired in sync with the locomotion bouts (Fig. 4e, f), and the maximal response was reached within 300 ms from the initiation of locomotion (Fig. 4i). Notably, the latency of the maximal response (i.e., maximum number of active CaMs) in OPC was about 3.6 s longer than the neuronal response (Fig. 4i; also see Fig. 2j – zoom-in of active bout).

To understand the anatomical interaction between OPCs and the cortical projections of LC neurons, we immunohistochemically visualized LC fibers expressing mGCaMP6s and OPCs in the S1 cortex of *Dbh-mGC6s* mice using antibodies against eGFP and PDGFRα, respectively (Fig. 4j, k). These experiments revealed a close contact between the LC fibers and OPC processes, and occasionally these cellular projections were intertwined with each other (Fig. 4k, inset II). Thus, OPCs are well placed to sense and respond to NE released by the LC fibers. To further ensure that activation of LC and concomitant NE release in response to locomotion indeed induces Ca²⁺ transients in OPCs, we specifically depleted the LC projections in the cortex of *Dbh-mGC6s* mice by injecting a single dose of the neurotoxin DSP4 (50 mg/kg)[49]. 3–4 days after DSP4 injection, we performed histological analysis to visualize LC fibers expressing mGCaMP6s, and found that almost two thirds (63.12%) of the LC projections in the cortex were successfully ablated (Fig. 4l, m). Next, we imaged Ca²⁺ transients in single OPCs in the S1 cortex of *NG2-GC6f;tdT* mice (Fig. 4n), then injected the mice with DSP4 (50 mg/kg). 3–4 days after the DSP-4 injection, we imaged Ca²⁺ transients in the same set of OPCs (Fig. 4o). As expected, the frequency of Ca²⁺ events in these OPCs originally increased in response to locomotion, but this effect was abolished in the same set of OPCs after the ablation of the LC fibers (Fig. 4p, q), and the A/R ratio diminished significantly (Fig. 4r). Next, we studied the properties of the Ca²⁺ transients in OPCs in the absence of locomotion-induced activation of Ca²⁺ signals in DSP-4 treated mice. We found that the frequency and the duration of locomotion-evoked Ca²⁺ events after the DSP-4 treatment were similar to that of the baseline Ca²⁺ events before treatment (Fig. 4p, q). However, there was a small increase in the amplitude of Ca²⁺ signals during locomotion in DSP-4 treated mice (Fig. 4u). These studies indicate that locomotion-induced increase in frequency of Ca²⁺ event in OPCs occurs through an action of NE, without major changes in the characteristics of Ca²⁺ signals.

It is possible that a long-term ablation of NE fibers can lead to increase in extracellular NE levels, which might affect neuronal activity and induce hyperactivity in mice[50]. To confirm that DSP-4 mediated ablation of NE fibers did not induce aberrant activity in the layer 2/3 neurons of the S1 cortex, we injected rAAVs expressing GCaMP6f

under control of human Synapsin promoter in the S1 cortex, and imaged Ca²⁺ transients in a stable population of neurons in vivo before and after the injection of DSP-4 (Supplementary Video 5, Supplementary Fig. 6a–i). We found that a similar proportion of L2/3 neurons were responsive to locomotion before (18%) and after (12%) DSP-4 injections (Supplementary Fig. 6j, k). These results indicate that the locomotion-induced activity in the layer 2/3 neurons of the S1 cortex is not a consequence of NE release, but largely occurs as a result of the mouse actively whisking and whiskers touching mHC walls during locomotion[51] (Supplementary Fig. 6–k). Additionally, we did not observe hyperactivity or any gross behavioral changes in the mice treated with DSP-4 (Supplementary Fig. 6l, m). These data further strengthen our observation that locomotion-induced Ca²⁺ transients in OPCs are primarily due to the activation of LC neurons and NE release, and not due to NE-mediated indirect activation of neurons adjacent to OPCs.

## Activation of alpha-adrenergic receptors on OPCs increases Ca²⁺ transients

To determine if OPCs express functional adrenergic receptors, and whether direct activation of these receptors can induce microdomain Ca²⁺ transients, we bath-applied a potent alpha1 adrenergic receptor (α1-AR) agonist, phenylephrine (10 μM; PE) in the presence of voltage gate sodium channel blocker TTX (0.5 μM), which blocks neuronal firing (Fig. 5a, b; Supplementary Video 6). We observed that activation of α1-AR on OPCs induced a large increase in the number of CaMs (3.3×), frequency (8.7×), amplitude (1.9×) and duration (1.9×) of Ca²⁺ events when compared to the baseline activity (in TTX, 0.5 μM) (Fig. 5e–g). To further validate that the increase in Ca²⁺ transients is due to the direct activation of α1-AR on OPCs (and not through an indirect activation of neurons, leading to the release of glutamate or GABA), we bath-applied PE in the presence of a cocktail of drugs composed of a neuronal activity blocker (TTX) combined with either (1) ionotropic and metabotropic glutamate receptors antagonists (Glut-B: CNQX, D-AP5, MCPG) (Figs. 5i–l) or (2) GABA (Gaba-B: Gabazine, CGP55845) receptors (Fig. 5m–p) antagonists. In the presence of these blockers, PE still evoked a large increase in the frequency and amplitude of Ca²⁺ events in OPCs (Fig. 5j–l, n–p), suggesting that OPCs express α1-ARs, and their activation can induce Ca²⁺ transients.

Since α1-ARs are also expressed by other non-neuronal cells, it is possible that the activation of these cells by NE can indirectly lead to an increase in Ca²⁺ transients in OPCs. Therefore, to confirm that indeed direct activation of α1-ARs in OPCs can mediate Ca²⁺ signaling, we isolated a pure OPC population using magnetic activated cell sorting (MACS) from the mouse cortex and cultured them for 2 weeks (Supplementary Fig. 7a, b). To simultaneously image OPC morphology and

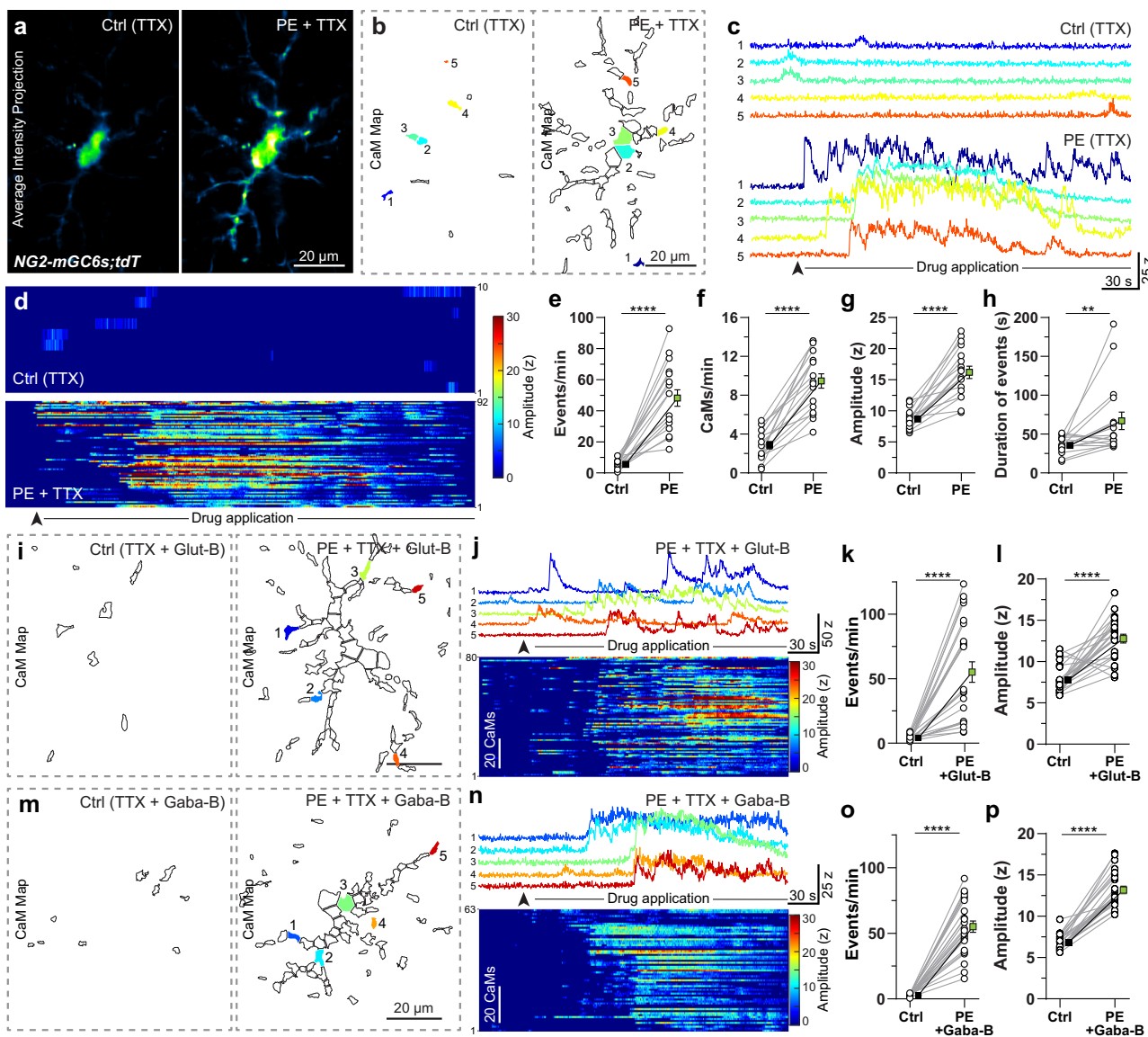

**Fig. 5 | Direct activation of α1 adrenergic receptors on OPCs induces Ca²⁺ transients. a** Pseudocolored average intensity projection of an mGCaMP6s-expressing OPC in an acute brain slice from a *NG2-mGC6s;tdT* mouse in control (Ctrl) conditions (TTX, 0.5 μM; left) and after bath application of phenylephrine (PE; 10 μM + TTX; 0.5 μM). **b** CaM map recorded in the Ctrl (left) and after PE application (right). **c** Intensity vs time Ca²⁺ traces from 5 CaMs in Ctrl (top) and after PE application (bottom) corresponding to colors in (**b**). **d** Heat-map of the intensity and temporal distribution of Ca²⁺ events in the Ctrl (top) and after PE application (bottom). Graphs showing changes in frequency of Ca²⁺ events (**e**), number of active CaMs per minute (**f**), average amplitude (z-scores) (**g**), and mean duration of Ca²⁺ events (**h**) in Ctrl and after PE application (**i**) Map of CaMs in control (Ctrl – Glut-B: TTX, 0.5 μM; CNQX, 10 μM; AP5,50 μM; MCPG, 10 μM) (left) and after PE + Glut-B application (right). (**j**) Intensity vs time Ca²⁺ traces from 5 CaMs after application of PE + Glut-B corresponding to colors in **i** (top). Heat-map of the intensity and

temporal distribution of Ca²⁺ events after PE + Glut-B application (bottom). **k**, **l** Graphs showing changes in frequency and average amplitude of Ca²⁺ events in Ctrl and after application of PE + Glut-B. **m** Map of CaMs in control (Ctrl – Gaba-B: TTX, 0.5 μM; CGP 55845, 5 μM; Gabazine, 5 μM) (left) and after PE + Gaba-B application (right). **n** Intensity vs time Ca²⁺ traces from 5 CaMs after PE + Gaba-B application corresponding to colors in **m** (top). Heat-map of the intensity and temporal distribution of Ca²⁺ events after PE + Gaba-B application (bottom). **o**, **p** Graphs showing changes in frequency and amplitude of Ca²⁺ events in Ctrl and after application of PE + Gaba-B. All data are presented as mean ± SEM. **e–h** $n = 17$ cells; $N = 4$ mice. **e–g** Paired t-test: ****$P < 0.0001$; **h** Wilcoxon matched-pairs rank test: **$P = 0.0011$; **k**, **l**, **o**, **p** $n = 24$ cells; $N = 3$ mice; **k**, **l**, **o** Paired t-test: ****$P < 0.0001$; **p** Wilcoxon matched-pairs rank test: ****$P < 0.0001$. Scale bars, 20 μm. Source data are provided in the Source Data file.

Ca²⁺ signals in vitro, we generated a novel fusion protein called Caprese – where we fused with a short linker green Ca²⁺ sensor jGCaMP8s[52] to red fluorescent protein mScarlet[53]. We derived expression of Caprese under a constitutively active ubiquitin promoter through a lentiviral vector, and imaged Ca²⁺ signals in OPCs (Supplementary Video 7). As in vivo, OPCs exhibit spontaneous Ca²⁺ transients mostly restricted to CaMs ($3.44 ± 2.73$ events/min; also see Supplementary Fig. 5k; Ctrl), with little to no Ca²⁺ activity in the cell soma (Supplementary Fig. 6c, d), confirming that Ca²⁺ signaling machinery is well preserved in OPCs

in vitro. Similar to the acute brain slice experiments, bath application of PE (10 μM) evoked robust Ca²⁺ transients in CaMs and the soma of OPCs (Supplementary Fig. 7c–h). In culture, we found that about 47% of PDGFRα + OPCs responded to PE, which is in line with our observation that not all OPCs respond to locomotion in vivo (see LNR-OPCs, Fig. 3). In addition, in these cultures, we could capture Ca²⁺ signals from MBP+ mature OLs which rarely exhibited spontaneous Ca²⁺ transients, and just 9% of OLs respond to PE (Supplementary Fig. 7i–o). We also performed 2-photon Ca²⁺ imaging on mature OLs in acute

brain slices from *Mog-mGC6s;tdT* mice and observed almost no baseline Ca$^{2+}$ transients, as was the case both in vivo and in vitro (Supplementary Fig. 7p–r). Bath-application of PE (+TTX, 0.5 μM) led to a prolonged increase in Ca$^{2+}$ trainsets in only 8 out of the 108 OLs imaged (Supplementary Fig. 7p–s). In summary, both the acute brain slices and in vitro experiments confirm that OPCs indeed express α1-ARs, and that their direct activation can engage G$_{\alpha q}$-GPCRs signaling, which lead to increase in intracellular Ca$^{2+}$ levels. Also, in line with our in vivo observation, we further confirmed that OPCs downregulated α1-ARs and stop responding to this mode of signaling upon differentiation into mature OLs.

## All three sub-type of alpha-adrenergic receptors are expressed on OPCs

There are 3 subtypes of α1-ARs, namely α1a, α1b and α1d, widely expressed in different brain areas[54]. All α1-ARs are coupled to G$_{\alpha q}$-type GPCRs and upon activation lead to cytosolic Ca$^{2+}$ increase through the opening of IP$_3$Rs on the endoplasmic reticulum (ER). To identify which α1-ARs subtypes are expressed by the OLCs, we performed simultaneous immunohistochemistry and single-RNA-molecule fluorescent in-situ hybridization (sm-FISH) in the S1 cortex of adult wildtype mice. We labeled OPCs, pmOL, and mature OLs using in-situ probes or antibody against *CSPG4* (NG2), *Enpp6*, and ASPA respectively. All three α1-AR subtypes were labeled with their respective in-situ probes, and nuclei of all cells were labeled with DAPI (Fig. 6a–c). Our quantitative analysis revealed that all three sub-types of α1-ARs are expressed by CSPG4+ OPCs, but at variable levels. Cells with >1 fluorescent puncta for a given α1-AR subtype in the close vicinity of their nucleus were considered positive for that α1-AR subtype (see "Methods"). Based on this criterion, approximately 59%, 47% and 62% of OPCs in S1 cortex expressed *Adra1a*, *Adra1b* and *Adra1d* α1-AR subtypes, respectively (Fig. 6d–f). As OPCs progressed towards a more differentiated stage, the expression of α1-AR subtypes was significantly reduced. For *Enpp6*+ pmOLs, only 7%, 22% and 23% of cells expressed *Adra1a*, *Adra1b* and *Adra1d* subtypes, respectively (Supplementary Fig. 8a–c, g). Once OPCs differentiated into ASPA+ mature OLs, the expression of *Adra1a, Adra1b* and *Adra1d* further reduced to 5%, 19% and 24% respectively (Supplementary Fig. 8d–g, h). This gene expression data is consistent with our observations that there is a diversity in the responsiveness of OPCs to locomotion in vivo and that not all OPCs respond to PE in acute brain slices and in culture. This also indicates that the significantly reduced response to locomotion or PE application by mature OLs is likely due to the downregulation of α1-ARs.

Next, to examine whether α1a, α1b and α1d AR subtypes are indeed involved in the enhancement of intracellular Ca$^{2+}$ in OPCs, we performed 2-photon Ca$^{2+}$ imaging in the S1 cortex of acute brain slices from 8-12 weeks old *NG2-mG6s;tdT* mice. Due to a lack of sub-type specific α1-AR agonists, we used sub-type specific α1-ARs antagonists to study the level of suppression of PE-evoked Ca$^{2+}$ transients in OPCs in the presence of these blockers. We found that the baseline Ca$^{2+}$ transients in OPCs persisted in the presence of TTX and the sub-type specific α1a (RS17035; 40 μM), α1b (Chloroethylclonidine, CEC; 30 μM) and α1d (BMY7378; 40 μM) antagonists when applied individually (Fig. 7g, k, o) or combined (Fig. 7s) suggesting that α1-ARs are not required for the generation of spontaneous Ca$^{2+}$ transients in OPCs in acute brain slices. When we bath-applied PE in the presence of α1a, α1b and α1d antagonists, OPC showed only 102%, 95%, and 33% increase in the frequency of microdomain Ca$^{2+}$ events respectively (Fig. 7i, m, q), with a slight increase in the amplitude of Ca$^{2+}$ events for α1b (23%), but not for α1a and α1d α1-ARs (Fig. 7j, n, r). These results are in stark contrast to the 866% increase in the frequency and 186% increase in amplitude of Ca$^{2+}$ events observed in OPCs when PE (+TTX) was bath-applied without α1-ARs antagonists (see, Fig. 5e, g). As expected, the application of PE in the presence of all three α1-ARs antagonists did not produce a significant increase in the frequency or amplitude of Ca$^{2+}$

transients in OPCs (Fig. 7s–v). In conclusion, these results further strengthen our conclusion that OPCs express all three subtype α1-ARs, which on activation can induce an intracellular Ca$^{2+}$ increase in CaMs.

## Direct activation of G$_{\alpha q}$-GPCR mediated Ca$^{2+}$ signaling in OPCs promote their differentiation into oligodendrocytes

Ca$^{2+}$ is thought to regulate all aspects of the OPC fate such as proliferation, differentiation and programed cell death[30]. To directly address the functional role of G$_{\alpha q}$-GPCR mediated Ca$^{2+}$ signals in regulating OPC fate, we took a cell-type specific chemogenetic approach. We used G$_{\alpha q}$ G-protein coupled hM3Dq Designer Receptors, which are Exclusively Activated by Designer Drugs (DREADD), and signal through phospholipase C mediated production of IP3, downstream activation of IP3 receptors (IP3Rs) and release of Ca$^{2+}$ from the internal stores[55]. At first, to confirm that indeed activation of hM3Dq DREADDs on OPCs increase intracellular Ca$^{2+}$ signals, we co-expressed hM3Dq-mCherry and GCaMP7s in primary OPCs cultures. Then, we performed 2-photon Ca$^{2+}$ imaging, and observed that a bath application of Clozapine N-Oxide (CNO, a designer drug to activate hM3Dq; 10 μM) enhanced the frequency of Ca$^{2+}$ transients 2-fold from the baseline (Ctrl:16.42 ± 5.65 events/min, and CNO: 32.78 ± 7.23 events/min) (Supplementary Fig. 9a–l; and Supplementary Video 8). Remarkably, the CNO evoked increase in the frequency of Ca$^{2+}$ transients were very similar to the Ca$^{2+}$ response seen by the activation of endogenous α1-ARs on OPCs (Supplementary Fig. 7g, h). Next, to express hM3Dq specifically in OPCs in the adult mouse brain and exogenously modulate Ca$^{2+}$ signaling in these cells, we generated a novel triple transgenic mice line. We cross-bred transgenic mice conditionally expressing hM3Dq tagged with a yellow fluorescent protein mCitrine under the control of CAG promoter[56] with NG2-CreER and tdTomato mouse lines (Fig. 7a). At 4 weeks of age, we injected the resulting DREADD mice (*NG2-CreER;CAG-LSL-hM3Dq-pta-mCitrine;Rosa26-LSL-tdTomato;* in short *NG2-hM3Dq;tdT*) and control (*NG2-CreER;Rosa26-LSL-tdTomato;* in short *NG2-tdT*) littermates with tamoxifen to induce the expression of hM3Dq-mCitrine and tdTomato in OPCs (Fig. 7b). 3 weeks post tamoxifen injections, histological analysis in the S1 cortex of the *NG2-hM3Dq;tdT* mice using antibodies against GFP and tdTomato to label hM3Dq-mCitrine and tdTomato expressing cells respectively, revealed OLC specific expression of these reporters (Fig. 7b), and nearly all (97.12 ± 0.90%) mCitrine+ cells co-expressed tdTomato (Supplementary Fig. 9m). 2 weeks after tamoxifen injection, we treated control and DREADD mice for 5 consecutive days with CNO (1 mg/kg). In addition, control and DREADD mice were exposed to the thymidine analogue BrdU throughout the CNO treatment to study the proliferation of OPCs in response exogenously enhanced Ca$^{2+}$ activity (Fig. 7a). Recent studies indicate CNO can be metabolized to clozapine, a serotonin and dopamine receptor antagonist, and induce effects which are independent of hM3Dq activation[57]. Therefore, to account for such off-target effects of CNO on the fate OPCs, we performed fate-mapping analysis on control mice injected with vehicle (0.9% saline: Sal) or CNO (Supplementary Fig. 9). In all experiments, we labeled control and hM3Dq expressing OPCs using antibodies against tdTomato and mCitrine respectively, and mapped the fate of OPC with the help of cell-fate markers such as PDGFRα (Fig. 7c, d), ASPA (Fig. 7e, f) and BrdU (Fig. 7g, h). Our fate mapping analysis in the S1 cortex of the *NG2-hM3Dq;tdT* and *NG2-tdT* mice revealed that enhancing Ca$^{2+}$ signaling in OPCs promoted their differentiation into mature OLs by twofold (Fig. 7i), and consequently depleted the hM3Dq+ OPCs pool (Fig. 7j). We also found that increased Ca$^{2+}$ signaling in hM3Dq+ OPCs suppressed their proliferation (Fig. 7k). As expected, we did not observe any overt effect of CNO administration on the proliferation and differentiation of OPCs (Supplementary Fig. 9n–p). Hence, based on these observations, we suggest that G$_{\alpha q}$-GPCR mediated Ca$^{2+}$ signaling in OPCs suppresses proliferation and guides their cellular fate towards the differentiation into mature OLs.

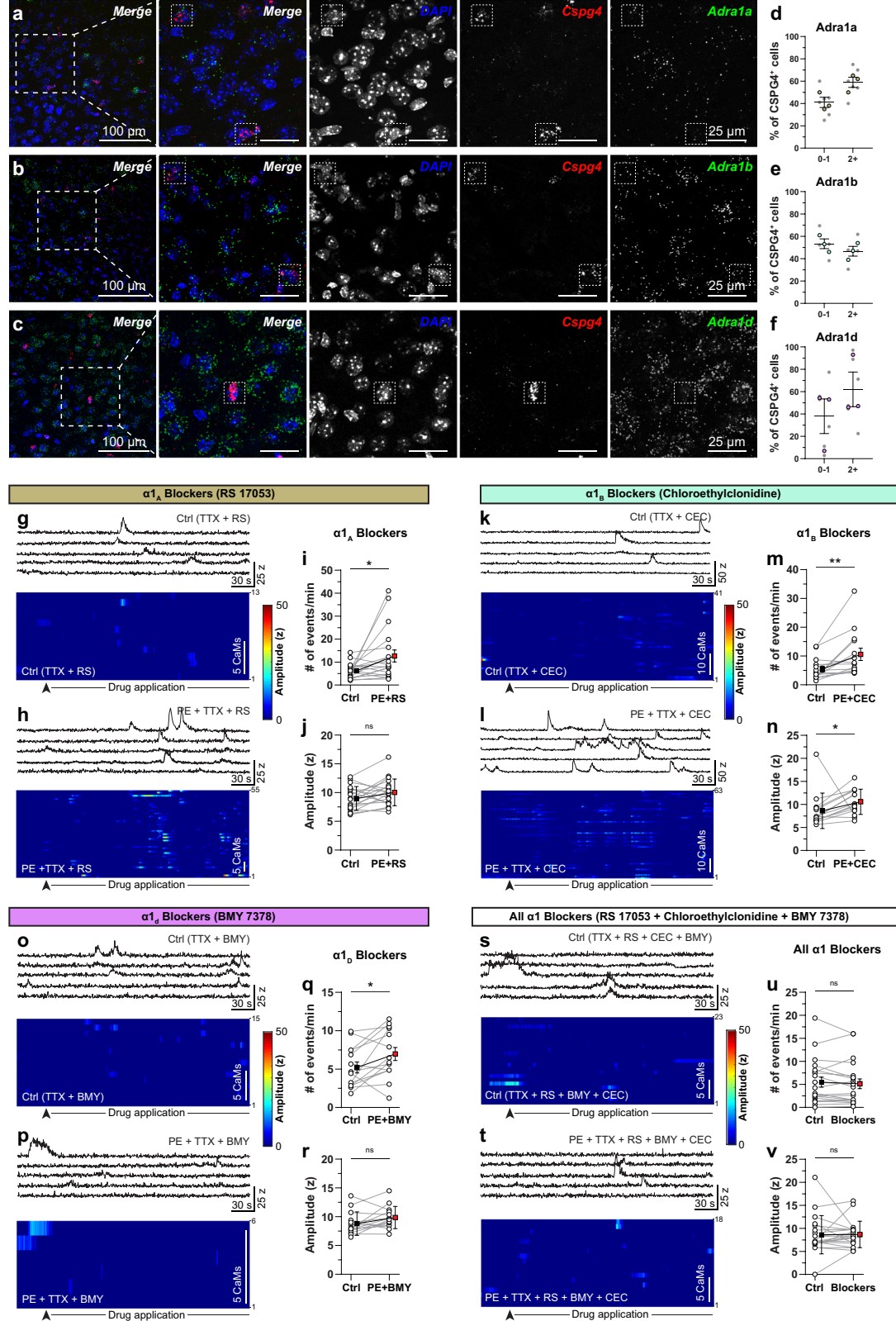

## NE suppresses proliferation and promotes differentiation of OPCs into oligodendrocytes

Norepinephrine is a potent neuromodulator and can regulate the proliferation and survival of neural progenitors[28,29]. To study whether NE-mediated $Ca^{2+}$ signaling in OPCs can directly modulate their fate, we took an in vitro approach. We isolated a pure population of cortical OPCs and cultured them for 2 weeks (Supplementary Fig. 10a–e). To study the effect of the activation of α1-ARs mediated $Ca^{2+}$ signaling in cultured OPCs, we added PE (10 μM) (Supplementary Fig. 10g), a mix of OPC differentiation-promoting thyroid hormones (T3 and T4 40 ng/mL) (Supplementary Fig. 10h), or PE and T3/T4 (Supplementary Fig. 10i) to the culture medium for one week, then performed a

**Fig. 6 | OPCs functionally express all three subtypes of α1 adrenergic receptors.** Maximum intensity projections of cortical brain sections labeled with a pan-nuclear maker DAPI (blue), OPC probe *Cspg4* (red), and probes for *Adra1a* (**a**), *Adra1b* (**b**) and *Adra1d* (**c**) adrenergic receptors (green). Graphs showing the percentage of Cspg4$^+$ cells that co-express Adra1a$^+$ (**d**), Adra1b$^+$ (**e**) and Adra1d$^+$ (**f**) puncta. **g–j** Pharmacological manipulation of OPCs in acute brain slices using the α1a antagonist RS 17053. Intensity vs time Ca$^{2+}$ traces from 5 CaMs in control (Ctrl − TTX, 0.5 μM and RS 17053, 40 μM) (**g**, top) and after PE + RS application (**h**, top). Heat-maps of the intensity and temporal distribution of Ca$^{2+}$ events in the Ctrl (**g**, bottom) and after PE + RS 17053 application (**h**, bottom). Graphs showing the frequency (**i**) and average amplitude of Ca$^{2+}$ events (**j**). The same experiments were also conducted using the α1b antagonist chloroethylclonidine (CEC) (Ctrl − TTX, 0.5 μM and CEC, 30 μM) (**k–n**), the α1d antagonist BMY7378 (Ctrl − TTX, 0.5 μM and BMY7378 10 μM) (**o-r**) or all 3 α1 antagonists simultaneously (Ctrl − TTX, 0.5 μM and all α1 blockers − RS, CEC, BMY) (**s-v**). All data are presented as mean ± SEM. (**d–f**) $n = 6$ sections, $N = 3$ mice; (**i, j**) $n = 19$ cells, $N = 3$ mice (**i**) Wilcoxon matched-pairs rank test: *$P = 0.0181$ and (**j**) Paired t-test: $P > 0.05$ (**j**); **m, n** $n = 14$ cells, $N = 5$ mice; Wilcoxon matched-pairs rank test: **$P = 0.0018$, *$P = 0.0166$; **q, r** $n = 14$ cells, $N = 5$ mice; **q** Paired t-test: *$P = 0.0303$, **r** Wilcoxon matched-pairs rank test: $P > 0.05$; **u, v** $n = 20$ cells, $N = 3$ mice; Wilcoxon matched-pairs rank test: $P > 0.05$. $P > 0.05$ not significant; ns. Scale bar, 100 μm. s seconds. z z-score. Source data are provided in the Source Data file.

detailed fate mapping analysis using immunocytochemistry to label OPCs (PDGFRα) and mature OLs (CNPase and MBP) (Supplementary Fig. 10f–i). With these experiments, we found that PE and a low concentration of T3/T4 modestly promoted the differentiation of OPCs by 22% and 20% respectively (Supplementary Fig. 10j, k). However, in the presence of T3/T4, PE enhanced the OPC differentiation by 34% (Supplementary Fig. 10j, k), indicating that activation of α1-ARs engages downstream signaling pathways which can have an additive effect on other differentiation-permissive factors. Next, to study whether NE-mediated Ca$^{2+}$ signaling in OPCs can modulate their fate in vivo, we chemogenetically activated LC neurons to induce NE release in the cortex (Fig. 8). We cross bred *Dbh-mGC6s* mice with a hM3Dq-mCitrine mouse line[56] to generate *Dbh-hM3Dq* (*Dbh-mGC6s;CAG-LSL-hM3Dq-pta-mCitrine*) (Fig. 8a). To study the effect of the activation of LC neurons on the fate of cortical OPCs, we treated control (*mGC6s;hM3Dq*) and Dbh-Gq (*Dbh-hM3Dq*) mice for 5 days with CNO (1 mg/kg). Along with CNO, we treated control and *Dbh-hM3Dq* mice with BrdU for two weeks to label proliferating OPCs (Fig. 8b). After the end of the BrdU treatment, we performed immunohistochemical analysis using antibodies against eGFP and cell-type specific markers for OPCs and OLs. As expected, the eGFP staining revealed LC fibers in the S1 cortex of *Dbh-hM3Dq* mice, but not in control mice (Fig. 8c–f), confirming that control mice don't express the G$_q$-DREADDs in LC neurons. Next, we quantified the number of ASPA+ OLs, PDGFRα + OPCs and BrdU+ proliferating cells in control and *Dbh-hM3Dq* mice (Fig. 8c–f). This analysis showed a 12.5% increase in the number of ASPA+ OLs (Fig. 8g) in *Dbh-hM3Dq* mice in which we exogenously activated NE fibers. Although we observed a trend towards reduction, there was no significant change in the density of BrdU+/PDGFRα + OPCs (Fig. 8h) in *Dbh-hM3Dq* mice. The density of PDGFRα + OPCs remained unchanged between the *control* and *Dbh-hM3Dq* mice (124.6 ± 5.24 cells/mm$^2$ and 125.9 ± 7.475, respectively) (Fig. 8i). We did not observe any increase in the number of BrdU+ OLs (<0.5 %), indicating that new OLs were not generated from the newly divided OPCs but from a pre-existing pool of OPCs. The percentage of BrdU+ cells which are OPCs also remained stable in *Dbh-hM3Dq* (52.78 ± 11.72%) when compared to the control (48.22 ± 11.38%) mice (Fig. 8j). In summary, results from these experiments support that NE-mediated signaling in OPC can promote their differentiation into OLs in vivo, without significantly affecting their proliferation.

## Discussion

OPCs represent the most abundant group of proliferating cells in the adult CNS. In vivo fate mapping studies indicate that they mainly serve as progenitors for OLs during development and adulthood, and contribute to the regeneration of OLs during demyelinating pathologies[58]. OPCs integrate several signaling pathways including growth factors and neuronal activity before they differentiate into myelinating OLs[15]. In this study, we discovered that OPCs and its lineage cells display a wide variety of Ca$^{2+}$ signals, and that these signals can be modulated by NE signaling, which in turn influences the fate of OPCs (Fig. 9).

### OPCs exhibit spontaneous and neuronal activity evoked microdomain Ca$^{2+}$ transients

Ca$^{2+}$ transients in OPCs were mainly seen in the processes and were restricted to micrometer-sized local hot-spots of activity (Fig. 2). Such localized hot-spots of cytosolic Ca$^{2+}$ signals, called Ca$^{2+}$ microdomains (CaMs), have previously been described in several cell types including neurons[59] and astrocytes[60]. In OPCs, somatic Ca$^{2+}$ transients were rarely seen and occurred only occasionally when mice engaged in intense locomotion activity (Fig. 2). Hence, unlike in neurons, the thresholds for summation of local CaM activity in processes to produce a cell-wide Ca$^{2+}$ response or somatic Ca$^{2+}$ transients are quite high in OPCs[61]. The source of microdomain Ca$^{2+}$ fluxes can be both extra- and intracellular, and as reported in excitatory neurons, CaMs in OPCs might exist in the absence of ultrastructural compartmentalization[59]. However, a detailed electron microscopic analysis will be required to characterize the ultrastructure of CaMs in OPCs. It is likely that, as reported in astrocytes, OPCs have two types of microdomain Ca$^{2+}$ transients – cell-extrinsic and cell-intrinsic[42]. Indeed, spontaneous OPC Ca$^{2+}$ activity in acute brain slices was reduced by -85% in comparison to baseline activity in vivo. In addition, a significant proportion of the baseline Ca$^{2+}$ activity in OPCs both in vivo and in acute brain slices (-25–35%) was dependent on the activation of ionotropic glutamate and GABA receptors (Supplementary Fig. 5). These results suggest that cell-extrinsic Ca$^{2+}$ events constitute the majority of Ca$^{2+}$ events in OPCs and are dependent on neuronal activity and the activation of neurotransmitter receptors. Additionally, we observed that the frequency of spontaneous Ca$^{2+}$ transients in pure OPCs in vitro (Supplementary Fig. 7) was similar to that of OPCs in acute brain slices exposed to a mix of neuronal activity blockers (3.44 events/min in culture; 2.85 events/min in acute brain slices), indicating that OPCs can generate neuronal activity independent cell-intrinsic Ca$^{2+}$ signals. Although defining the sources of such cell-intrinsic spontaneous Ca$^{2+}$ events would necessitate further investigation, these signals could be generated via transient activation of store-operated Ca$^{2+}$ entry, mechanotransduction channels, reactive oxygen species (ROS), and Ca$^{2+}$ efflux from mitochondria[42,62–64].

Whether excitatory post-synaptic currents (EPSCs) at OPC-axon synapses generate enough Ca$^{2+}$ flux at OPC CaMs to be visualized as 'fast' Ca$^{2+}$ transients remains an open question[65]. Of all the Ca$^{2+}$ signals we recorded in OPCs, about 45% of Ca$^{2+}$ signals in OPCs CaMs were of sub-second duration, about -1.5% of all Ca$^{2+}$ events were as 'fast' as 330 ms, and might represent EPSCs at OPC-axon synapses (Supplementary Fig. 3). Both in vivo and in acute brain slices, glutamate and GABA receptor antagonists reduced the frequency of baseline Ca$^{2+}$ transients in OPCs, indicating that OPCs respond to the neurotransmitters released at synapses (Supplementary Fig. 5). Although subcellular targeted GCaMP6f has been shown to report EPSC-mediated Ca$^{2+}$ fluxes in cultured neurons[66], in vivo we could not accurately parse out ionotropic and metabotropic Ca$^{2+}$ events in OPCs based on the temporal speed (duration) of these events estimated with cytosolic GCaMP6f. Thus, we suggest that in OPCs 'fast' Ca$^{2+}$ signals (330–600 ms) could be a mix of opening of calcium permeable AMPA (cpAMPA) and NMDA receptors, A-type voltage gated Ca$^{2+}$ channels and even activation of various G$_{\alpha q}$-GPCR coupled metabotropic receptors[41,61].

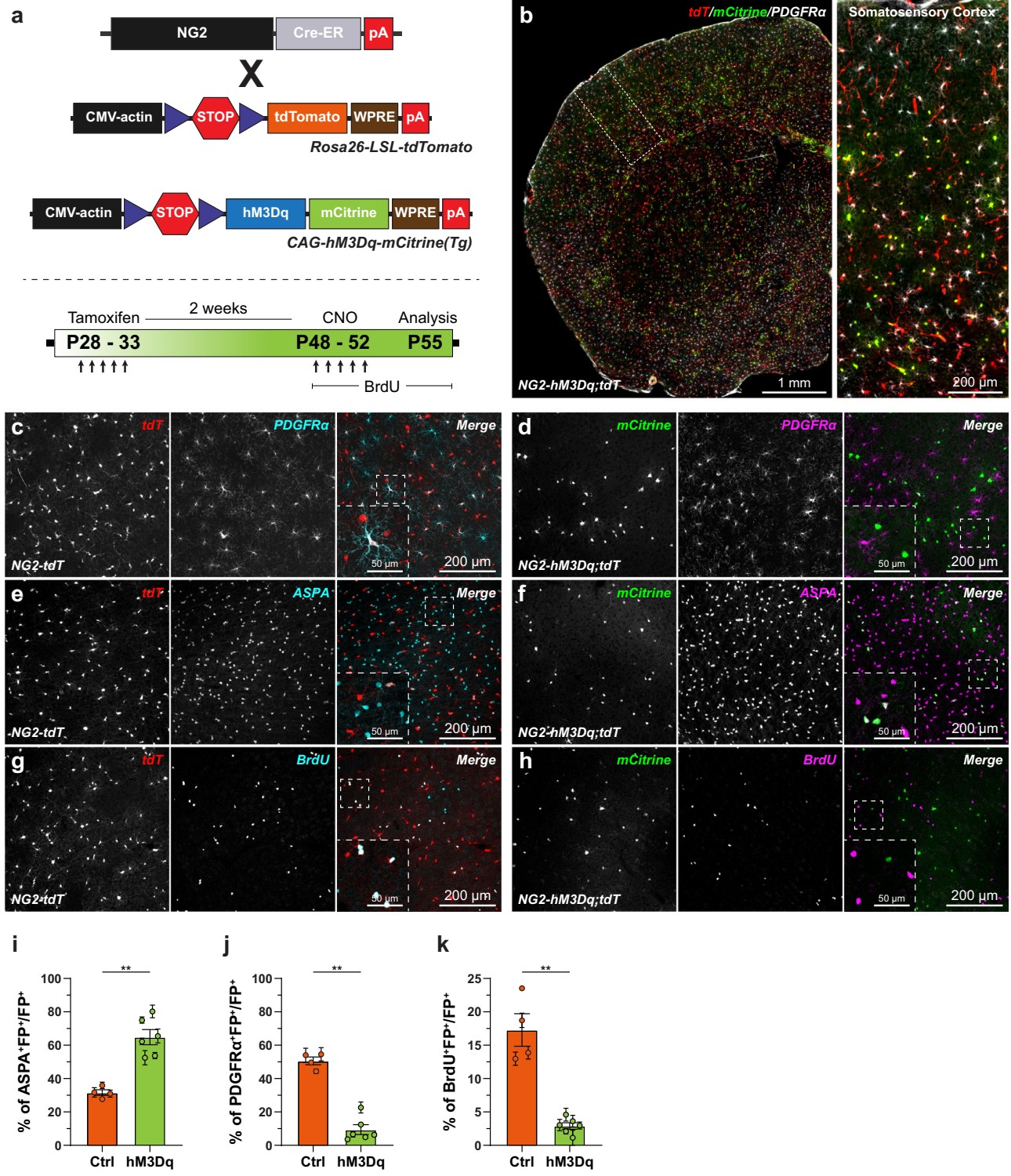

**Fig. 7 | Chemogenetic activation of Ca²⁺ signals in OPCs enhances differentiation and suppresses proliferation. a** Cartoon showing a transgenic strategy to express tdT and a chemogenetic effector hM3Dq in OPCs of *NG2-hM3Dq;tdT* mice (top) and the strategy used to chemogenetically activate Ca²⁺ increase in OPCs. 6-7 weeks old mice were injected with CNO for 5 consecutive days and BrdU for one week to label actively dividing cells (bottom). **b** (left) Representative example of a coronal brain sections showing the distribution of tdT⁺ (red), hM3Dq⁺ (mCitrine, green) and PDGFRα⁺ (grey) cells in the brain. (right) Zoom-in image from the S1 cortex (boxed area on left). **c–h** Maximum intensity projections of confocal z-stacks of the coronal brain sections co-stained with tdT (red; **a**, **e**, **g**), mCitrine (green; **d**, **f**, **h**), PDGFRα (**c**, **d**), ASPA (**e**, **f**) and BrdU (**g**, **h**) (cyan) antibodies. (Inset) show high-magnification images of the boxed area. Graphs showing the percentage of FP+ (tdT or mCitrine) cells that were ASPA+ (**i**), PDGFRα⁺ (**j**), and BrdU+ (**k**) in the S1 cortex of control (Ctrl) or *Dbh-hM3Dq* (hM3Dq) mice. All data are presented as mean ± SEM. **i–k** *n* = 12 sections, *N* = 4 (Ctrl); *n* = 18 sections, *N* = 6 mice (hM3Dq); Mann–Whitney test: **P* = 0.0095. Scale bars, 1 mm and 200 μm (**b**); 200 μm and 50 μm (inset) (**c–h**). Source data are provided in the Source Data file.

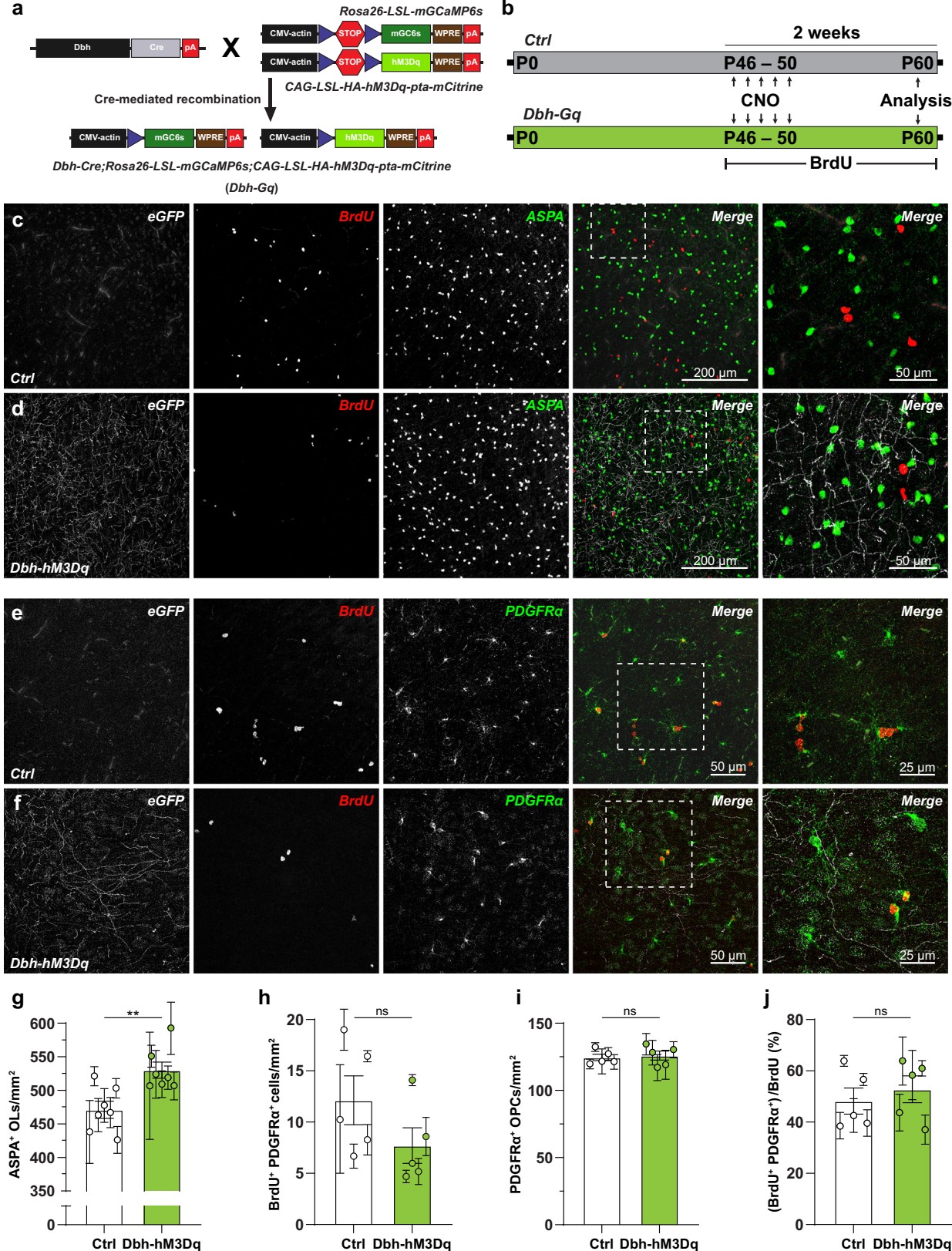

## Relationship between microdomain Ca²⁺ signals and fate of OPCs

The in vivo fate mapping and Ca²⁺ imaging studies revealed that OPCs exhibit distinct Ca²⁺ dynamics while undergoing fateful decision of cell division or differentiation (Fig. 3). All OPCs we imaged had similar CaM activity at baseline, but a small population (~28%) of them did not respond to locomotion (LNR OPCs). This is the pool of OPCs in which proliferation occurred. After cell-division, the summation of the number of CaMs in the two daughter cells was equal to those in the mother OPC (Fig. 9). In addition, like the mother cell, the newly born daughter cells did not respond to locomotion (Supplementary Fig. 2). This highlights that OPCs undergoing cell-division distribute synapses

**Fig. 8 | Norepinephrine regulates fate of OPCs by enhancing differentiation.** **a** Cartoon showing a transgenic strategy to express mGCaMP6s and a chemogenetic effector hM3Dq in noradrenergic (NA) neurons projecting to the S1 cortex in *Ctrl* (Control; *mG6s;hM3Dq*) and *Dbh-hM3Dq* mice. **b** Strategy to chronically activate NA neurons and induce NE release. 6–7 weeks old *Ctrl* and *Dbh-hM3Dq* mice were injected with CNO for 5 consecutive days to induce NE release, and mice were treated with BrdU for two weeks to label actively diving cells. **c, d** Maximum intensity projections of confocal z-stacks of the coronal brain sections immunostained stained for eGFP (grey), BrdU (red; cell proliferation marker) and ASPA (green; mature oligodendrocyte marker) from *Ctrl* (**c**) and *Dbh-hM3Dq* (**d**) mice.

(right) High-magnification image of the boxed areas in (**c, d**). **e, f** Maximum intensity projections of confocal z-stacks of the coronal brain sections immunostained stained for eGFP (grey), BrdU (red) and PDGFRα (green; OPC marker) from *Ctrl* (**e**) and *Dbh-hM3Dq* (**f**) mice. (right) High-magnification image of the boxed areas in (**e, f**). **g–i** Graphs showing the density of ASPA+ OLs (**g**), BrdU$^+$/PDGFRα$^+$ OPCs (**h**), PDGFRα$^+$ OPCs (**i**), and (**j**) percentage of BrdU+/PDGFRα + OPCs in control and *Dbh-hM3Dq* mice. All data are presented as mean ± SEM. **g** $n = 21$ sections, $N = 7$ mice; Mann−Whitney test: **$P = 0.0070$. **h–j** $n = 15$ sections, $N = 5$ mice; Mann−Whitney test: $P > 0.05$. Scale bars, 200 μm and 50 μm (boxed areas). ns, not significant. Source data are provided in the Source Data file.

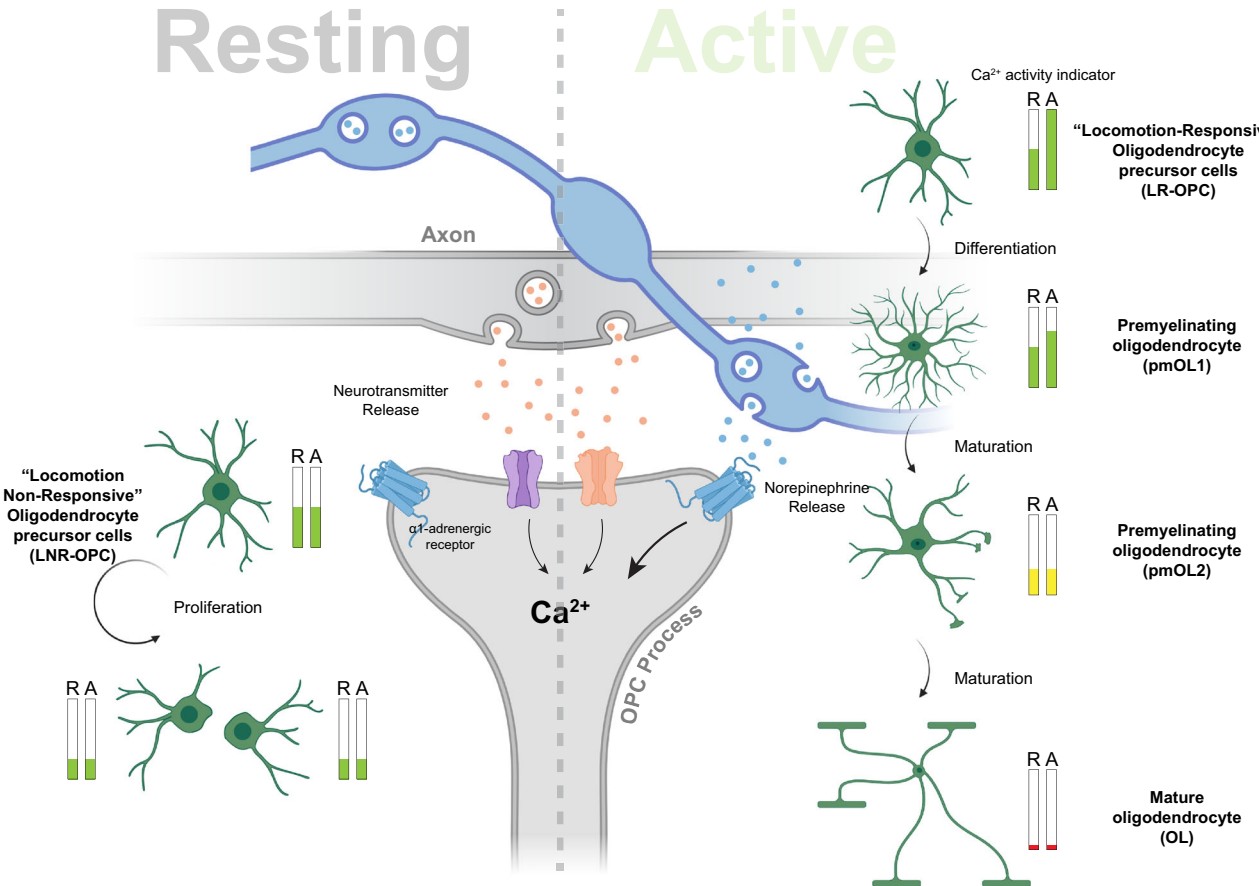

**Fig. 9 | Oligodendrocyte lineage cells display distinct baseline and norepinephrine-evoked Ca$^{2+}$ signals during proliferation and differentiation.** OPCs sense neurotransmitters (e.g., Glutamate, GABA) and neuromodulators (NE) at the OPC-axon junctions. During the resting phase (R: when mouse did not engage in exploratory behavior), the Ca$^{2+}$ activity of oligodendrocyte lineage cells (OPCs, pmOL1, pmOL2 and OLs) is mainly driven by the baseline neuronal activity in the cortex. As OPCs differentiate into pmOL2 and OL the baseline activity is significantly reduced (see Ca$^{2+}$ activity indicator bars). During arousal, when mouse is alert and engages into exploratory bouts of locomotion (A: active phase), NE is released from cortical projections of LC neurons, which enhanced the Ca$^{2+}$ activity

in OPCs and pmOL1, but not in pmOL2 an OLs. About 30% of OPCs do not show increased Ca$^{2+}$ activity in response to locomotion and preferentially proliferate – we called these OPCs Locomotion Non-Responsive (LNR-OPCs). A newly born daughter OPCs has half the baseline activity of the mother cell, and also does not respond to locomotion (see Resting side, left). OPC population which responded to exploratory locomotion bouts by increasing frequency of Ca$^{2+}$ transients are more likely to differentiate into pmOLs and later mature into OLs - we called them Locomotion Responsive OPCs (LR-OPCs). Chemogenetic activation LC-neurons to release NE or a direct increase in OPCs Ca$^{2+}$ signals promoted the differentiation of OPCs into mature OLs. Part of the illustration was created using BioRender.

and the repertoire of neurotransmitter receptors equally amongst the two daughter cells—a cell-division feature unique to OPCs[44,67]. Thus, LNR OPCs and newly generated OPCs possibly maintains synaptic contacts with unmyelinated axons, and express ionotropic and metabotropic receptors for glutamate and GABA, while expressing little-to-no neuromodulator receptors such as α1-ARs (Fig. 3). Several studies suggest that activation of AMPA and GABA receptors in OPCs promote OPC proliferation[68], and that neuromodulator such as NE suppresses proliferation of neural progenitors[26]. It is likely LNR OPCs maintain low

expression of α1-ARs to keep them free from the cell-proliferation brakes of NE[31].

Recent studies tried to classify OLCs into various sub-classes based on their unique mRNA signatures[23] and electrophysiological properties[43]. One such study combined single-cell RNA sequencing and in vivo Ca$^{2+}$ imaging techniques to characterize spinal cord (SC) OPCs in larval zebrafish, and reported the existence of two distinct pools of OPCs based on the anatomical location and Ca$^{2+}$ fluctuations[31]. OPCs with high baseline Ca$^{2+}$ activity were located in neuronal soma-

rich areas and those with low baseline Ca$^{2+}$ activity were located in axo-dendritic rich areas of the SC. This study reported that OPCs with low baseline Ca$^{2+}$ activity (in axon-rich areas) preferentially differentiated into OLs. While we identified two pools of OPCs within the cortical grey matter in the brain of adult mice, unlike in the zebrafish SC, both the OPC pools exhibited similar level of baseline activity[31]. However, both mouse OPC types exhibited distinct response to neuromodulatory signals i.e., one sub-type (LN OPCs) responded to locomotion and other sub-type (LNR OPCs) did not (Fig. 9). We found that LN OPCs respond to NE and tend to differentiate into OLs. Interestingly, a recent study showed that axo-dendritic area in zebrafish SC receive dense arborization of noradrenergic fibers[69]. It is therefore possible that OPCs in zebrafish SC also integrate NE-mediated signaling to regulate their differentiation into OLs. Our findings will prompt further studies to investigate the role of neuromodulatory cues in regulating fate of OPCs during development, in distinct regions of the brain and SC, and across different model systems.

Another study, entirely based on single-cell transcriptomics, described that committed OPCs (COPs) and early stage pmOLs (likely pmOL1 in this study) had higher expression of α1-ARs and IP$_3$R2[23], features that may endow these cells with a capacity to sense and respond to NE mediated G$_{\alpha q}$-GPCR signaling. Hence, we suggest those COPs and pmOLs, which can effectively integrate neuromodulatory cues and exhibit enhanced Ca$^{2+}$ signaling, have a higher probability to mature into OLs. Indeed, when we chemogenetically increased G$_{\alpha q}$-GPCR and IP3Rs mediated Ca$^{2+}$ signaling in OPCs in adult brain, they differentiated into mature OLs, and very few OPCs with enhanced Ca$^{2+}$ signals proliferated (Fig. 7). Hence, we conclude that, in the adult brain, a baseline synaptic activity at OPC-axon junctions keep OPCs in a proliferative state[70], while promoting GPCR-mediated Ca$^{2+}$ signaling through neuromodulator action might guide the fate of OPCs towards differentiation. In addition to fate determination, could Ca$^{2+}$ signals have other roles in pmOLs? Two more developmental studies on larval zebra fish suggest that a modest Ca$^{2+}$ rise in premature OLs triggers myelin sheath elongation, whereas high amplitude, long duration Ca$^{2+}$ events induce sheath shortening and retractions events[32,33]. This indicates that pmOLs at later developmental stages (likely pmOL2 in this study) integrate information encoded in localized Ca$^{2+}$ transients to trigger myelin formation and fine-tune myelin internodes. However, further detailed analysis will be required to understand the role of pmOLs Ca$^{2+}$ transients in regulating myelination in the adult brain. Remarkably, as OLs become fully mature, they undergo major physiological changes such as fluctuations in membrane potentials[18], activation of K$^+$ leak currents, expression of TRPA1 channels[71] and downregulation of several G$_q$-coupled metabotropic receptors[23]. These changes potentially translate into the very low baseline Ca$^{2+}$ transients, and the loss of locomotion evoked and NE-mediated Ca$^{2+}$ signaling we observed (Fig. 3, Fig. 9 and Supplementary Fig. 4), suggesting that this mode of Ca$^{2+}$ signaling doesn't play essential role in OL function.

## OPCs express adrenergic receptors and their activation induces microdomain Ca$^{2+}$ transients

It well-established that locomotion induces NE release by activating adrenergic neurons in the LC, which promotes states of vigilance and wakefulness in mice[72]. In this study, we uncovered that locomotion-induced microdomain Ca$^{2+}$ signals in OPCs is due to the activation of α1-ARs on OPCs. Several observations implied that the locomotion-induced Ca$^{2+}$ signals in OPCs could be evoked by the action of NE (Fig. 4) – (1) during an active exploration (i.e. locomotion) mice are in the state of vigilance, and NE is released by the Dbh$^+$ neurons in LC, which project throughout out the brain[46]; (2) axonal projection from Dbh$^+$ neurons criss-cross the entire cortex and make a close contact with OPC processes; and (3) ablation of LC fibers in the cortex, using the neurotoxin DSP-4[49], abolished locomotion-induced activation of

CaMs in OPCs. Glutamatergic neurons release neurotransmitters within a few milliseconds once they are depolarized[73]. Hence, if glutamate or GABA release at the OPC-axon synapses would lead to locomotion induced Ca$^{2+}$ transients, OPCs would have responded instantaneously. However, we observed a delay of about 4 s for OPCs to reach the maximal activation of CaMs from the instant the mouse engaged in locomotion (Fig. 4). It takes about 1.87 s for Dbh$^+$ neurons to release NE from the time they were depolarized[73], and the half-life of NE in the brain is about 3.1 s[74], thus we infer that Ca$^{2+}$ transients in OPCs in response to locomotion are perhaps driven by neuromodulators such as NE. Also, blocking ionotropic glutamate and GABA receptors in vivo and in acute brain slices did not prevent locomotion evoked or α1-ARs agonist induced Ca$^{2+}$ signals, respectively (Supplementary Fig. 5).

Previous studies using transgenic mice expressing eGFP reporters under the control of promoters for α1a and α1b ARs showed that α1a ARs are expressed in OPCs but not in mature OLs[75], and α1b ARs are expressed in both OPCs and OLs[76]. Although α1d-AR promoter-driven LacZ reporter expression shows these receptors are abundant in the cortex, a cell-type specific expression analysis has not yet been performed[77]. A recent study on single-cell RNA sequencing of OLCs showed that α1a and α1b ARs were expressed in various OLCs and that α1b-ARs was the most expressed sub-type[23]. However, none of these studies showed that α1-ARs are functionally active on OPCs, or explored their dynamic expression in OLCs. Our systematic single-RNA molecule fluorescent in-situ hybridization (smFISH), Ca$^{2+}$ imaging and extensive α1-ARs pharmacology showed that about 50% of OPCs expressed various α1-AR sub-types, and that this proportion decreases as they differentiate into pmOL and OL (Fig. 5 and Supplementary Figs. 7, 8). In this study, we performed exhaustive analysis across different preparations (in vivo, ex vivo and in vitro) to confirm that OPCs indeed express α1-ARs and that their direct activation induces Ca$^{2+}$ signals. Yet, there remains a possibility that some of the effect of NE on OPCs can be due to an indirect action of neurotransmitters and growth factors released by activation of other neural cells including neurons and astrocytes[46,54] by NE. In the future, availability of more efficient OPC-specific Cre-driver mouse lines for gene deletion, and floxed mice for conditional deletion of Adra1b and Adra1d genes will allow us to perform a systematic deletion of all three α1-ARs in OPCs, and enable us to investigate the more definitive role of cell-autonomous α1-ARs mediated Ca$^{2+}$ signaling in OPCs. In this study, we nonetheless highlight the role of neuromodulators such as NE in regulating Ca$^{2+}$ signals and cell-fate of OPCs in adult brain in vivo, and shed light on distinct mechanisms for regulating OPC fate in an activity-dependent manner.

## Norepinephrine regulates fate of OPCs and promotes differentiation

Despite the fact that all OPCs have axon-OPC synapses and that the synaptic activity at these junctions is known to guide their fate, it remains unknown why very few OPCs successfully differentiate, mature and myelinate in the adult brain[3]. NE might be the missing link between the basal neuronal activity sensed by OPC and their fate. A brain wide volumetric release of NE inhibits random and spontaneous neuronal activity, strengthens neuronal connectivity, and potentiates sensory and motor functions[78]. Low NE levels are associated with sedation or sleep, whereas high levels of NE lead to arousal. A transcriptomics and histological analysis performed in mice during sleep and wake cycles suggested that the proliferation rate of OPCs doubled during sleep whereas wakefulness promoted the differentiation of OPCs[79]. Like neurons, we propose that NE enhances the gain of ongoing baseline activity at the axon-OPC synapses, thereby promoting the differentiation and maturation of OPCs into myelinating OLs. In line with this hypothesis, we observed that NE promoted differentiation of OPCs both in vitro and in vivo (Fig. 8 and Supplementary Fig. 10). It is now becoming evident that there are two modes of

myelination – (1) neuronal activity independent and, (2) activity-dependent, which requires the presence of permissive growth factors and neuronal activity[15,80,81]. In our culture experiments, we observed that activation of NE signaling in OPCs further enhanced the differentiation promoting effect of thyroid hormone (an OPC differentiation factor)[82]. This suggests that neuromodulators such as NE might promote the maturation of OPCs which are already primed to become differentiated.

We also made an unusual observation - in general, OPC differentiation is associated with increased OPC proliferation events[83]. Instead, we observed no change in the proliferation of OPCs in vivo in response to chemogenetically enhanced NE signaling (Fig. 8). This observation may have exposed another exciting feature of the effect of NE on the development of the oligodendrocyte lineage. OPCs are known to express both α and β ARs[23,24], which means NE could possibly work on two levels – (1) NE mediated activation of β-ARs suppresses proliferation and promotes differentiation of OPCs[84] and (2) as OPCs progressed towards a more differentiated state (COPs and pmOLs) the activation of α1-ARs further promotes their maturation into myelinating OLs. It is well established that pmOLs are susceptible to death and that only a few survive in vivo[3,85]. We suggest that IP$_3$R2+/α1-AR+ pmOLs, which integrate neuromodulatory cues, are more likely to mature into OLs[23]. Thus, pmOLs might represent a promising pool of cells which effectively respond to NE signaling, and contribute to the adaptive myelination and in myelin repair[86,87].

Although very little is known about the role of NE signaling on OPCs in vivo, reduced levels of NE and hypertrophy of TH+ neurons in the LC were reported in the brains of individuals with multiple sclerosis (MS)[88]. Similar observations were made in the cortex and SC of a mouse model of demyelination (experimental autoimmune encephalomyelitis, EAE), which had reduced NE levels[89,90]. In addition, studies on such mouse models indicate that increased levels of NE in the CNS can reduce disease severity[88,91]. NE could also be a connecting link between myelin and OLCs dysfunction seen during stress and depression[92]. For example, serotonin and norepinephrine reuptake inhibitors (SNRIs) such as venlafaxine, which enhance NE levels in the brain, promote myelination and improve cognitive function in mouse model demyelination[93]. Thus, enhanced NE levels through SNRIs or electrical stimulation of LC neuron e.g., using tDCS (transcranial direct current stimulation) might have the potential for improved remyelination in MS[94].

In summary, we have uncovered that OLCs exhibit a wide variety of Ca$^{2+}$ signals, which can be modulated by NE. NE signaling plays a pivotal role in regulating survival, proliferation and differentiation of OLC (Fig. 9). Future studies will help us clarify the mechanisms by which this NE signaling contributes to adaptive myelination and myelin plasticity in adult brain, and explore the therapeutic potential of this mode of signaling to promote remyelination and myelin repair in demyelinating diseases.

## Methods

Both male and female mice were used for all experiments, and mice were randomly allocated to experimental groups. For ex vivo experiments, adult mice aged 6–12 weeks old were used and for in vivo experiments, mice aged 8–16 weeks old were used unless otherwise described. Mice were maintained on a 12 h light/dark cycle, at a temperature of 22 ± 2 °C and a humidity level between 50–60%. Food and water were provided ad libitum. Animal studies were approved by the Governmental Council Karlsruhe, Germany (G-126/20 and G-189/20). All animal experiments were carried out in a strict compliance with German Animal Protection Law (TierSCHG) at the Heidelberg University, Germany.

### Generation of ROSA26 targeted conditional mGCaMP6s reporter mouse line
To localize GCaMP6s to the plasma membrane, we fused the gene sequence encoding the first 8 amino acids of the modified MARCKS

sequence (MGCCFSKT) to the first methionine (i.e. start ATG) of GCaMP6s sequence (termed mGCaMP6s). To enhance expression of mGCaMP6s, we used a strong ubiquitous CMV-βactin hybrid (CAG) promoter and placed the woodchuck hepatitis virus posttranscriptional regulatory element (WPRE) at the 3′ end of mGCaMP6s expression sequence. For inducible expression of mGCaMP6s, a loxP flanked STOP cassette was placed upstream of the mGCaMP6s (Supplementary Fig. 1a). The mGCaMP6s transgenic construct was targeted to the ubiquitously expressed ROSA26 locus using homologous recombination in mouse embryonic stem (ES) cells derived from a SV129 mouse strain. Transgenic mice were generated at the Johns Hopkins University Transgenic Core Laboratory[95]. Germ line transmission was achieved by breeding male chimeric founders to C57Bl/6 N wild-type female mice.

### Transgenic mice
Generation and genotyping of CreER driver lines NG2-CreER[35] (Jax# 008538), Dbh-Cre[48], GCaMP6f[96], tdTomato reporter mouse lines[36] (Jax# 007914), Mogi-Cre[45] and hM3Dq-Cirtine chemogenetic effector mice[56] (Jax# 026220) have been previously described (Supplementary Methods Table 1).

### Tamoxifen Injections
The tamoxifen solution for injections (10 mg/mL) was prepared by dissolving tamoxifen (Sigma, T5648) in sunflower seed oil (Sigma, S5007). To ensure the tamoxifen was fully dissolved, the solution was sonicated at room temperature for approximately 5 min, then vortex for 30 s. This process was repeated until total dissolution was achieved and no crystals remained visible in the solution. The solution was then stored at 4 °C, protected from light, for a maximum of 5 days. Mice aged 4–6 weeks were injected intraperitoneally (i.p.) with tamoxifen at a dosage of 100 mg/kg of body weight once a day for 3 or 5 days, depending on the experiments. Each injection was performed at least 20 h after the previous one. All subsequent experiments were performed at least 2 weeks after the last tamoxifen injection.

### CNO and BrdU treatment
Clozapine-N-Oxide (CNO) solution was prepared by dissolving water-soluble CNO (HelloBio, HB6149) in 0.9% saline at a concentration of 100 μg/mL. Mice were injected with CNO at a dosage of 1.0 mg/kg (i.p.) daily for 5 consecutive days. During the same period, mice were given 5-Bromo-2′-deoxyuridine (BrdU) (Sigma, B5002). BrdU was dissolved in 0.9% saline at a concentration of 10 mg/mL. Mice were injected twice daily with BrdU at a dosage of 50 mg/kg (i.p.) for 5 consecutive days. For maximum labeling of proliferating cells, beginning on the day of the first BrdU injections, mice were also provided with a 1.0 mg/mL BrdU in 1% sucrose solution as drinking water.

### Norepinephrine fiber ablation
To deplete noradrenergic fibers, mice were injected with the noradrenergic neuron specific neurotoxin N-(2-chloroethyl)-N-ethyl-2-bromobenzylamine hydrochloride (DSP4; Sigma, C8417) once at a dose of 50 mg/kg (i.p.). To ensure NE fibers were successfully depleted, subsequent histological and Ca$^{2+}$ imaging experiments were carried out at least 3 days after the DSP-4 injection.

### Immunohistochemistry
Mice were deeply anesthetized by injecting pentobarbital (Narcoren) at 150 mg/kg (i.p.), and transcardially perfused with 4% paraformaldehyde (PFA) in 0.1 M sodium phosphate buffer (pH 7.4). Brains were carefully dissected out from the cranium and post-fixed in 4% PFA for 16 h (overnight) at 4 °C. The brains were then transferred to 1× phosphate-buffered saline (PBS) and 35 μm thick coronal sections were cut using a vibratome (Leica, VT 1000 s). Sections were collected in PBS containing 0.2% sodium azide and stored at 4 °C until further

processed. Prior to immunostaining, the sections were permeabilized 0.3% Triton X-100 in 1× PBS for 10 min at room temperature (RT). To prevent non-specific binding of antibodies, the sections were then placed in a blocking buffer (0.3% Triton X-100 and 5% normal donkey serum (NDS) in 1× PBS) for 1 hr at RT. Then, the sections were incubated overnight with the primary antibodies (Supplementary Methods Table 2) diluted in blocking buffer at 4 °C with gentle shaking on an orbital shaker. Sections were rinsed 3 times for 5 min each in 1× PBS at RT, and incubated for 3 h at RT with a blue fluorescent nuclear dye 4′,6-diamidino-2-phenylindole (DAPI) and fluorescent dye-conjugated secondary antibodies (Supplementary Methods Table 3) diluted in blocking buffer. Finally, sections were rinsed thrice for 5 min each in 1x PBS at RT. Sections were then mounted on SuperFrost Plus slides (Thermo Fisher Scientific, 4951PLUS) using Aqua Polymount (Polyscience Inc, 18606-20) and stored at 4 °C to dry for at least 16 h prior to imaging. Stained coronal sections were then imaged using an epifluorescence microscope (Leica DM6000) and automatically stitched in the accompanying software (LAS X, Leica) upon acquisition. Images were further processed and analyzed using ImageJ. For fate mapping analysis, we quantified cells from all 6 layers of the S1 cortex, and represented the data as the mean of three sections/mouse, for 5–7 mice per genotype.

### Single RNA-molecule fluorescent in situ hybridisation (sm-FISH)

8 weeks old C57BL/6N mice were transcardially perfused with 4% PFA and their brains were isolated and post-fixed as described in the previous section (Immunohistochemistry). Following the post-fixation overnight in 4% PFA at 4 °C, the brains were dehydrated by sequentially storing them in 10%, 20% and 30% sucrose in 1× PBS at 4 °C and until they sank to the bottom of the vial in each solution. Brains were then cut to fit a 7 mm × 7 mm × 5 mm (L × W × H) embedding mold, frozen in optimal cutting temperature compound (OCT, Tissue-Tek) with dry ice and stored at −80 °C. The tissue was later sliced into 16 μm thick sections using a cryostat (Leica, CM1950), and stored in an airtight container at −80 °C until further processed. In situ hybridizations was performed using the RNAscope Multiplex Fluorescent Kit v2 (Advanced Cell Diagnostics) for fixed frozen tissue according to the manufacturer's instructions with the following probes: Mm-Adra1a-C2 (408611-C2), Mm-Adra1b-C2 (413561-C2), Mm-Adra1d (563571), Mm-Cspg4-C3 (404131-C3), and Mm-Enpp6-C4 (511021-C4). Fluorescent signals were then amplified using a TSA Plus Fluorescein kit (PerkinElmer, NEL741E001KT). In addition, some sections were stained with DAPI and anti-ASPA antibodies (see Immunohistochemistry). All sections were mounted on SuperFrost Plus slides (Thermo Fisher Scientific, 4951PLUS), and later imaged using a confocal microscope (see Confocal Imaging).

smFISH confocal image stacks were imported into Image J for quantification. The nuclei of oligodendrocyte lineage cells were identified using a combination of DAPI and a cell-type specific marker (Cspg4, Enpp6, or ASPA), then carefully outlined. Puncta were manually counted on a plane-by-plane basis from the confocal z-stacks for optimal precision. Cells were only considered positive if the puncta were located directly on the nucleus or its perimeter.

### Confocal imaging

Images were acquired using a Leica SP8 Confocal microscope equipped with either a 40× (HC PL APO CS2 Water-immersion, Leica) or 63× (HCX PL APO Oil immersion, Leica) objectives, with the pinhole set to 1 Airy Unit (AU). Images were acquired at a resolution of 1024 × 1024 pixels, with a system-optimized step size of 0.423 μm. Image size varied between 290.62 × 290.62 μm and 87.87 × 87.87 μm depending on the objective and magnification factor used. Images were then further processed using ImageJ, and presented as pseudocolored maximum intensity projections.

### Morphological analysis of OLCs

The morphology of OLCs was analyzed using the tdTomato channel on either in vivo 2-photon images (average intensity time-series projection of an entire recording) or confocal microscopy z-stack images. Importantly, confocal z-stacks were limited to only a few frames to ensure that the resulting analyzed image could be reliably compared to the 2-photon average time-series projections. Then, using ImageJ, we manually segmented the soma and traced the skeleton of each cell with the Simple Neurite Tracer (SNT)[97] plugin for ImageJ and imported them to MATLAB using a custom code. We then computed the area of the soma and its circularity using the MATLAB *regionprops* built-in function. Next, we calculated the distances between consecutive main branches as a proportion of the circumference of the soma and used standard deviation of these distances to estimate the spread of main processes around the OLCs soma

### Acute brain slice preparation

Mice were anesthetized with isoflurane and immediately decapitated with a pair of large scissors. Their brains were quickly dissected out and stuck on a 5 mm agar stage before being mounted on a vibratome (Leica VT1200S) equipped with a sapphire blade. 250 μm thick coronal cortical slices were cut in ice-cold N-methyl-D-glucamine (NMDG) based cutting solution containing (in mM): 135 NMDG, 1 KCl, 1.2 $KH_2PO_4$, 1.5 $MgCl_2$, 0.5 $CaCl_2$, 10 Dextrose, 20 Choline Bicarbonate (pH 7.4 and ~310 mOsm/L). Cortical slices were then incubated in artificial cerebrospinal fluid (ACSF) containing (in mM): 119 NaCl, 2.5 KCl, 2.5 $CaCl_2$, 1.3 $MgCl_2$, 1 $NaH_2PO_4$, 26.2 $NaHCO_3$ and 11 Dextrose (292–298 mOsm/L) for 30 min at 37 °C, then maintained at RT for the entire duration of the experiment. The NMDG cutting solution and the ACSF were continuously bubbled with carbogen (95% $O_2$/5% $CO_2$). All slice imaging experiments were performed at the RT.

### Pharmacological manipulations

All pharmacological experiments on the acute brain slices or cultured cells were performed by dissolving drugs directly in ACSF, and applying them through a fast perfusion system. Unless otherwise specified, all experiments were performed in the presence of a voltage-gated $Na^+$ channel blocker TTX (0.5 μM, HelloBio HB1035) to block neuronal firing of action potentials. Following drugs were used for pharmacological manipulations (Supplementary Methods Table 4): adrenergic α1 receptor agonist – phenylephrine (10 μM Sigma P6126); AMPA & Kainate receptors antagonist – CNQX (10 μM, HelloBio HB0205); NMDA receptors antagonist (D-AP5 (50 μM, HelloBio HB0225); Group I & II metabotropic glutamate receptors antagonist – (S)-MCPG (10 μM, HelloBio HB6112); $GABA_A$ receptor antagonist – Gabazine (5 μM, HelloBio HB0901); $GABA_B$ receptor antagonist– CGP55845 (5 μM; HelloBio HB0960); adrenergic $α1_A$ receptor antagonist–RS17053 (40 μM, Tocris 0985); adrenergic $α1_B$ receptor antagonist –Chloroethylclonidine (30 μM, Sigma-Aldrich B003); and adrenergic $α1_D$ receptors antagonist–BMY 7378 (10 μM, Tocris 1006).

### Time-lapse two-photon microscopy of ex vivo and in vitro preparations

Intracellular $Ca^{2+}$ concentration changes and morphology of OLCs were imaged on a custom-built 2-photon microscope (Bergamo II, ThorLabs) fitted with a 20× water-immersion objective (1.0NA, XLUMPLFLN, Olympus), 8 kHz galvo/resonance scanner, piezo drive for fast z-scanning, and two GaAsP PMT detectors. 2-photon excitation for GCaMP6f, mGCaMP6s, jGCaMP8s and tdTomato at a single wavelength was achieved using a mode locked Ti:Sapphire pulsed laser (Chameleon Ultra II, Coherent) tuned at 940 nm. Images were acquired at a resolution of 512 × 512 pixels (85.5 × 85.5 μm) and 12-bit pixel depth, with each imaging session consisting of a 5-min recording of 920 frames at 3.1 Hz. Over the entire course of the imaging, the acute brain slices/cover slips were steadily perfused with oxygenated ASCF. Since

it was difficult to identify OPCs expressing mGCaMP6s in acute brain slices, they were briefly exposed to 10 μM phenylephrine prior to each experiment, and responsive cells with OPC morphology were further imaged.

To validate the identity of cells in culture, after Ca²⁺ imaging the cover slips containing cultured OPCs were fixed in 4% PFA for 15 min at RT and rinsed 3 times with 1X PBS. Antigen retrieval was then performed by incubating coverslips in L.A.B solution (Polysciences 24310) for 10 min. We then performed immunohistochemical staining (see Immunohistochemistry for brain slices) with markers for OPCs and mature OLs on these cover slips to locate each imaged OLC and identify their cell-type (e.g. OPC, pmOL or OL).

## OPC culture
Brain cortices from 4–5 days old C57BL/6N pups were dissociated into single cell suspensions using Neural Tissue Dissociation Kit (T) (Miltenyi Biotec) for magnetic activated cell-sorting (MACS) according to the manufacturer's protocol. In order to eliminate astrocytes and obtain an enriched fraction of OPCs, cells were isolated from cortical suspensions by 2-step MACS during which anti-GLAST (ACSA-1) and anti-O4 microbeads (Miltenyi Biotec) were used sequentially. Isolated OPCs were seeded on poly-D-lysine (Gibco A3890401) coated glass coverslips at an approximate density of 30,000 cells/cm². Cells were kept in OPC medium composed of DMEM/F12 medium (Gibco 11330032) supplemented with SM1 (StemCell Technologies 05711), N2B (StemCell Technologies 07156), Penn/Strep (50 U/mL, Thermo-Fisher 15140), N-Acetyl-Cysteine (5 mg/mL, Sigma-Aldrich A8199), Forskolin (5 mM, Calbiochem 344270), CNTF (5 mg/mL, PreproTech 450-13), PDGF-AA (20 ng/mL, PreproTech 100-13A), and NT-3 (1 ng/mL, PreproTech 450-03B). The medium was changed every other day until the cells reached optimal confluency of 60–80% for further experiments.

## Viral vector transduction
We produced lentiviruses by co-transfecting GCaMP7s or Ubc-Caprese (cytoplasmic jGCaMP8s with an N-terminus fusion of mScarlet red fluorescent protein) expression vector with three packaging vectors (pRSV, pMDL and pVSVG) into HEK293 cells using polyethylenimine (PEI, Polysciences 23966-1). The virus-containing media was then collected 24 h and 48 h post-transfection, and ultra-centrifuged at 25300 rpm for 2 h at 4 °C (SW 32 Ti Swinging-Bucket). The pelleted viral particles were resuspended in OptiMEM medium (Gibco, 31985-047), aliquoted and stored at −80 °C until use. Custom-made mCherry-hM3Dq (pLV[Exp]-mCherry-UBC>hM3Dq) lentiviruses were purchased from VectorBuilder. One week (6–7 days) after the start of the culture, OPCs were transduced with the appropriate viruses. The Ca²⁺ imaging experiments were performed 2–3 days post transduction (see Time-lapse two-photon microscopy of ex vivo and in vitro preparations).

## In vitro OPC fate mapping
Once the cover slips with OPCs reached 60–70% confluence, PDGF-AA was removed from culture medium. 48 h later, we began differentiating the OPC under four different conditions: (1) Control (no additional component); (2) 10 mM PE; (3) 40 ng/mL tri-iodo-thyronine (T3, Sigma-Aldrich T6397) + 40 ng/mL thyroxine (T4, Sigma-Aldrich T1775); or (4) 10 mM PE + 40 ng/mL T3 + 40 ng/mL T4. Reagents were freshly supplemented every day to the cell culture medium. One week later, the cells were fixed, stained, imaged and quantified (using the protocol described in **Immunohistochemistry** for brain slices) to evaluate and compare the efficiency of each differentiation protocol.

## Glass cranial windows implantation
Mice aged 5–8 weeks old were implanted with chronic glass cranial windows[42,98]. Mice were first anesthetized using isoflurane (5% for induction, 1.5–2% during surgery, mixed with 0.5 L/min O₂), and placed on a custom stereotaxic frame (RWD) fitted with a stereo microscope (Stemi 508, Zeiss) and thermostat-controlled heating pad (RWD). Throughout the surgery, mice were maintained at 37 °C body temperature, and were administered dexamethasone (2 mg/kg, i.p.) and 0.03 mg/kg buprenorphine sub-cutaneously (s.c.) to reduce inflammation and pain, respectively. Skin on the scalp was removed to expose the skull over the right cerebral hemisphere, where a 2 × 2 mm area (centered at 1 mm below Bregma and 3 mm lateral from the midline) was marked using a scalpel. This perimeter was gently drilled to remove the bone, and a 2 × 2 mm piece of cover glass (#1, Menzel-Gläser) was placed within the craniotomy and sealed to the skull using histoacrylamine glue (Histoacryl, Braun). A helicopter headbar with a 5.2 mm circular opening (model 2, NeuroTar) was attached using dental cement (Kulzer, Paladur) mixed with cyanoacryl (Hager Werken, 152261). After the surgery, mice were administered carprofen (5 mg/kg, s.c.) every 12 h for two days to reduce pain and inflammation. Whenever possible, multiple mice were housed in the same cage with 1–2 wooden shelters, providing them with an enriched environment. Mice were allowed to recover from surgery for two weeks, before Mobile HomeCage training could begin.

## AAV Injection in Somatosensory Cortex
C57BL/6 N mice aged 6–8 weeks were obtained from Janvier Labs and anesthetized by i.p. injection of a mixture of medetomidine (Sedin, 1 mg/mL, Vetpharma Animal Health), midazolam (5 mg/mL, Hameln Pharma GmbH) and fentanyl (0.05 mg/mL, Piramal Critical Care) in a 2:3:8 ratio (110 μL/injection). Mice were then placed on a custom stereotaxic frame and were maintained at 37 °C body temperature (see previous section). Skin was gently removed to expose the skull and the injection site (1 mm below Bregma and 3 mm lateral from the midline) was marked and drilled. Then, a borosilicate glass capillary (World Precision Instruments, Kwik-Fil, 1B100-4) with a tip diameter of ~100 μm was front-filled with the ssAAV-9/2-hSyn1-ch1-GCaMP6f-WPRE-SV40p(A) virus (ZNZ Neuroscience Centre Zurich, Viral Vector Facility, v83-9, titer: 5.5 × 10¹² vg/mL). 300–500 nL of the virus were injected 0.65 mm below the dorsal surface of the somatosensory cortex (see coordinates above) at a rate of 20 nL/min (Micro-4, Microsyringe Pump Controller, World Precision Instruments). To ensure optimal delivery of the virus to the targeted site, the pipette was withdrawn over 10 min after the injection was completed. Following the cortical virus injection, we implanted a cranial window over the injection site following the protocol in the above section. Finally, the anaesthesia was antagonized by i.p. injection of a mixture of atipazole (5 mg/mL, Prodivet Pharmaceuticals), flumazenil (0.1 mg/mL, Fresenius Kabi) and naloxone (0.4 mg/mL, Inresa Arzneimittel GmbH) in a 1:1:6 ratio. After the surgery, mice were administered carprofen (5 mg/kg, s.c.) every 12 h for two days to reduce pain and inflammation.

## Mobile HomeCage setup and training
All intravital 2-photon imaging sessions in awake animals were performed using a high precision 2D locomotion tracking device (Mobile HomeCage (mHC), NeuroTar). The mHC consists of a solid aluminum platform coupled to an air compressor which allows a constant upwards flow of air through holes on its surface. The uniform air-cushion it creates is sufficient to support an ultralight carbon-fiber cage (180 mm diameter, 70 mm height) with negligible friction. When mice were placed inside, they were able to easily displace the carbon-fiber cage and engage in unrestricted self-motivated exploration while still being head-fixed, enabling us to study Ca²⁺ dynamics in OLCs and mouse behavior simultaneously. In addition, high-precision magnetic trackers embedded in the mHC platform constantly calculate the position and the speed of the animal inside the experimental setup using two small magnets located at the bottom of the carbon cage.

To ensure mice are comfortable being head-fixed in the mHC, mice were progressively acclimatized over the course of one week to

remain head-fixed for about 60 min during awake 2-photon microscopy sessions in the mHC. On the first two days, the mice were introduced to the 2-photon imaging environment and extensively handled by the experimenter. The handling steps included occasional manual head-restriction to simulate imaging conditions inside the mHC. On the third day, mice were placed for ~15 min inside the carbon-fiber cage of mHC floated on the air-cushion, and could freely explore the imaging environment without head-fixation. Finally, over the next 4 days, the mice were head-fixed in mHC and placed under the 2-photon microscope for an increasing period of time (+10 min per day), replicating the exact conditions of an imaging session. By the end of the 7th day of habituation, mice were fully habituated and minimally stressed while head-fixated in mHC, and were ready for imaging session lasting between 45–60 min.

### Intravital two-photon microscopy on awake behaving mice

As previously mentioned, intravital 2-photon microscopy in awake mice was performed using the mHC setup, enabling us to correlate $Ca^{2+}$ dynamics in OLCs with explorative behavior of mice. To ensure synchronization of behavioral and $Ca^{2+}$ imaging data, image acquisition on the 2-photon microscope was initiated by a TTL pulse from the Mobile HomeCage motion tracking software (Tracker version 2.1.10.3). Images were acquired on a custom-built 2-photon microscope (Bergamo II, ThorLabs) with a 20× water-immersion objective (1.0NA, XLUMPLFLN, Olympus). 2-photon excitation was achieved using a mode locked Ti:Sapphire pulsed laser (Chameleon Ultra II, Coherent) tuned at 940 nm (see Time-lapse two-photon microscopy in acute brain slices section for more details). Unless otherwise specified, image time-series were acquired at a rate of 5.1 Hz and a resolution of 512 × 512 pixels (85.5 × 85.5 μm) for a total of 10 min. Mice were kept head-fixed in the mHC for a maximum of one hour per imaging session.

### In vivo pharmacology

Following a one-hour pre-drug 2-photon imaging session, the mice were removed from the mHC and injected i.p. with a combination of glutamatergic (AMPA: NBQX 10 mg/kg, NMDA: CGP 33551 10 mg/kg) and GABAergic (GABA$_A$ Bicuculline 4 mg/kg) ionotropic receptor antagonists. Shortly after the injection (~40 min), the mice were placed back under the 2-photon microscope for another 1-h imaging session and the same cells imaged during pre-drug session were located and re-imaged.

### Mobile HomeCage data analysis

All data about the location and speed of mice in the Mobile HomeCage were imported to MATLAB (Mathworks, 2021a) using a custom code. Active periods were detected automatically and aligned with the 2-photon imaging data using a custom MATLAB code (Mathworks, 2021a). Locomotion bouts were defined as the period during which the speed of the mouse was above threshold, which was set at one standard deviation above the mean speed of the animal during that session. Importantly, some mice displayed low levels of motor activity, significantly lowering the mean speed and standard deviation used to establish the threshold, which could lead to pseudo-bouts bouts being detected. Hence, bouts were excluded if they did not contain speeds above 20% of that of the maximum observed in that session. In addition, to account for residual locomotion-induced $Ca^{2+}$ activity that occurs immediately after the end of an active phase, each locomotion bout was extended by ~2.5 s. This delay corresponds to an interval larger than 90% of interspike distances observed during locomotion, which would ensure that events occurring within that period would be correctly classified as locomotion-induced rather than included in the resting events. Mice were considered to be in a resting state when the speed remained below threshold for an uninterrupted period of at least 30 s. $Ca^{2+}$ signals in OLCs were sorted into Resting or Active categories corresponding to the type of bout during which these events were initiated.

### Preprocessing of image stack for $Ca^{2+}$ signal analysis

To correct for movement artefacts in XY plane, 2-photon time-series image stacks were registered post hoc using automatic alignment algorithm TurboReg[99] (plugin for ImageJ). During registration, TurboReg transforms images into a 32-bit format. Therefore, after registration, images were reformatted to 16-bit. Surrounding background pixels that were not part of the cell (based on median-intensity projection of the image stack) including pericytes and larger blood vessels were cropped to accelerate analysis. For an automatic analysis of $Ca^{2+}$ microdomain and $Ca^{2+}$ signals in OLCs, our machine learning based algorithm CaSCaDe was modified accordingly for specific imaging datasets (see details below).

### Extraction and analysis of $Ca^{2+}$ signal in oligodendrocyte lineage cells

Using the average projection of the 2-photon image stack, we automatically generated a $Ca^{2+}$ microdomain map as described in our prior publication[42]. Prior to event detection, the traces from each $Ca^{2+}$ microdomain were extracted and flattened using the MATLAB *detrend* function. Since awake in vivo imaging can lead to movement-induced changes in fluorescence that are independent of the $Ca^{2+}$ concentration, we also extracted fluorescence signals from the static tdTomato channel using the same $Ca^{2+}$ microdomain map. Because the fluorescence of tdTomato does not change as a function of $Ca^{2+}$ concentration, any rapid changes in fluorescence observed in this channel can be interpreted as an artifact. To avoid including such artifacts in our analysis, we thus subtracted the tdTomato traces from those of GCaMP6f for microdomains. $Ca^{2+}$ events were automatically detected using peak detection algorithm of CaSCaDe. Only $Ca^{2+}$ events with amplitude more than 5 z-scores and duration at least 4 frames were considered. For each $Ca^{2+}$ event, the beginning and ending points were determined using the point of maxima for each peak as a reference. Beginning of the peak was defined as the last point preceding the maximum that has a value below 1 z-score, and the ending is defined as a point where 1 z-score is reached for two consecutive frames, or falls within 10% of the starting point's value. The resulting $Ca^{2+}$ signals were classified based on our previously trained SVM classifier[42].

### Fast imaging (15 Hz)

To acquire image stack at a 15 Hz sampling frequency, the imaging was performed at 2 frame averaging and 256 × 256 pixels resolution, which resulted in higher pixel sizes (0.17 μm/pixel at 5 Hz vs 0.33 μm/pixel at 15 Hz) and increased background noise. Hence, CaSCaDe was modified as follows to accommodate images with lower signal to noise ratio. First, a filtering step was added to exclude inactive or highly noisy microdomains that did not exhibit clear (7+ z-score) $Ca^{2+}$ events. Next, peak detection was performed on the selected microdomains as described in the previous section. Since images acquired at 15 Hz are inherently noisy, an accurate estimation of the duration of $Ca^{2+}$ event was not possible based on the method implemented to analyze images acquired at 5 Hz. To resolve this, the microdomain $Ca^{2+}$ traces were passed through a 1 Hz lowpass filter. Then, start and end points of $Ca^{2+}$ events were computed separately for both the original unfiltered and filtered traces. This method allowed for optimal detection of both fast events (using the unfiltered traces) and slow events (using the filtered traces) simultaneously. Hence, a combination of both binary detection signals (unfiltered and filtered) was computed to obtain the final duration of each event. Importantly, hyper-segmented events (i.e., events that were detected separately in the unfiltered trace, but that correspond to a single event in the filtered trace) were combined if they occurred less than 0.2 s apart. The resulting $Ca^{2+}$ signals were also classified based on our SVM classifier[42].

## Neuronal Ca²⁺ analysis (15 Hz)

Neuronal Ca²⁺ time-series were acquired at a 15 Hz sampling frequency and 256 × 256 pixels resolution, as described in the Fast imaging (15 Hz) section above. To correct for movement artefacts in XY plane, 2-photon time-series image stacks were registered post hoc using the Flow Registration[100] plugin for ImageJ. Using the resulting average intensity projection, we manually traced ROIs around the soma of every neuron present in the field of view, which were then used as input for CaSCaDe to extract the Ca²⁺ dynamics. Due to the rapid firing pattern of neurons, CaSCaDe would occasionally underestimate the number of Ca²⁺ signals if they occurred in quick succession. To circumvent this issue, the raw Ca²⁺ traces were also transformed (Δf/f) and used as an input in a spike inference algorithm from CASCADE (Global_EXC_15Hz_smoothing200ms)[101]. We combined this spike probability output (at a threshold of 0.2) with our own CaSCaDe output, allowing us to adequately segment consecutive peaks. Neurons were considered responsive if they produced a Ca²⁺ transient within 4 seconds of the initiation of the locomotion bout in at least 50% of the bouts.

## Noradrenergic fibers Ca²⁺ analysis

Due to high noise levels, the automatic segmentation on CaSCaDe resulted in spurious microdomain maps with hypersegmentation of ROIs. Using the average projection of the imaging stacks, noradrenergic fibers were manually identified and regions of interest (ROIs) delineated. Manual ROIs were imported into MATLAB as binary masks and used as a template for CaSCaDe to generate the Ca²⁺ microdomain map and to detect Ca²⁺ events.

## Statistics and reproducibility

Statistical analyses were performed using GraphPad Prism (version 9.4.0). Each dataset was individually tested to assess the normality of its distribution using the D'Agostino-Pearson normality test. If both datasets passed the normality test, a two-tailed t-test was used to determine the statistical significance of the difference between two groups. Otherwise, a non-parametric test (Wilcoxon-matched pairs signed rank test) was used for paired comparisons. If three or more datasets were compared, we used one-way ANOVA with Tukey's multiple comparisons test or Kruskall–Wallis test with Dunn's correction to verify statistical significance between groups. All reported $p$-values used for statistical analysis were derived from two-tailed tests. The statistical tests used for each comparison (and their significance level) are indicated in the figure legends. Data are shown in the figures as mean ± SEM and $p < 0.05$ was considered significant. $N$ represents the number of animals and $n$ the number of cells (physiology) or brains slices (histology).

The images presented throughout the manuscript are taken directly from the dataset they are meant to represent, and the number of replicates for each experiment is indicated in the corresponding figures legends. In cases where qualitative characterization of the mouse lines (Fig. 1, Supplementary Data Fig. 1) and other immunohistological stainings (Fig. 4b, j, k) were performed, the experiments were repeated a minimum of 3 technical and 3 biological replicates (i.e., $N = 3$ mice).

## Data presentation

All the main and supplemental figure panels were assembled in Adobe Illustrator 2023, and cartoons were modified from graphical elements available at BioRender.

## Reporting summary

Further information on research design is available in the Nature Portfolio Reporting Summary linked to this article.

## Data availability

All data generated or analyzed during this study are included in this published article and its supplementary information files. Source data are provided with this paper. All other data are available from the lead author upon request (A.A., amit.agarwal@uni-heidelberg.de). Source data are provided with this paper.

## Code availability

The code used for our Ca²⁺ data analysis software CaSCaDe has already been published[42] (https://ars.els-cdn.com/content/image/1-s2.0-S0896627316310078-mmc10.pdf). The modifications to the code are detailed in the Methods section, and are available from the lead author upon request (A.A., amit.agarwal@uni-heidelberg.de).

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

## Acknowledgements

We thank Dr. Dwight Bergles (Johns Hopkins University, Baltimore, USA) for generously sharing *Rosa26-lsl-mGCaMP6s*, which were generated by A.A. while he was a post-doctoral fellow in the laboratory of Dr. Bergles. We thank Dr. Akiko Nishiyama (Department of Physiology and Neurobiology, University of Connecticut, USA) and Dr. Ari Waisman (Institute for Molecular Medicine, Mainz, Germany) for kindly sharing *NG2-CreER* and *Mogi-Cre* mouse lines. We thank Chiara Olmeo and Dr. Claudio Acuna Goycolea at the Institute for Anatomy and Cell Biology, Heidelberg for providing GCaMP7s lenti viruses. Also, we want to thank Dr. Thomas Kuner, Department of Functional Neuroanatomy, Institute for Anatomy and Cell Biology, Heidelberg, for the institutional support. We acknowledge the data storage service SDS@hd supported by the Ministry of Science, Research and the Arts Baden-Württemberg (MWK), and

the German Research Foundation (DFG) through grant INST 35/1314–1 FUGG and INST 35/1503–1 FUGG. F.F. is a graduate student at the Heidelberg Biosciences International Graduate School (HBIGS), and we thank HBIGS for their support. A.A. was supported in part by the Chica and Heinz Schaller Research Foundation, and grants from the Deutsche Forschungsgemeinschaft: FOR 2289 (P8; AG 287/1-1) and SFB1158 (A09). A.A. acknowledges a partial financial support towards the publication fee by Deutsche Forschungsgemeinschaft within the funding program "Open Access Publikationskosten" as well as by Heidelberg University.

## Author contributions

F.F. conducted awake in vivo, in vitro and acute brain slice $Ca^{2+}$ imaging experiments, performed fate mapping studies, contributed to the development of imaging analysis algorithms, and analyzed data. K.A. developed and adapted algorithms for analysis of $Ca^{2+}$ imaging and Mobile HomeCage data, and analysed $Ca^{2+}$ imaging and cell-morphology data. R.R.D characterized transgenic mouse lines used for $Ca^{2+}$ imaging and chemogenetics experiments, contributed in the cell culture and in vivo fate mapping experiments, performed sm-FISH based gene expression analysis. F.B. performed in vitro $Ca^{2+}$ imaging and fate-mapping experiments. R.R.D and F.B. contributed equally to the work. I.C. performed histological characterization of Dbh-Cre mice and injected the AAVs for neuronal imaging. A.H. contributed to the initial characterization of transgenic mice used for $Ca^{2+}$ imaging and performed preliminary pharmacological experiments in acute brain slices. A.A. conceived the project, designed the experiments, generated and characterized Rosa26-lsl-mGCaMP6s mice, acquired the funding and supervised the project. A.A. and F.F. wrote the manuscript with help from the other authors. All authors have read and approved the manuscript.

## Funding

## Competing interests

The authors declare no competing interests.
