## [Peer Review File · Nature Communications]

Norepinephrine regulates calcium signals and fate of oligodendrocyte precursor cells in the mouse cerebral cortexEditorial Note: This manuscript has been previously reviewed at another journal that is not operating a transparent peer review scheme. This document only contains reviewer comments and rebuttal letters for versions considered at *Nature Communications*.

REVIEWERS' COMMENTS

Reviewer #1 (Remarks to the Author):

The revision of the transferred manuscript is exceptional - the manuscript is significantly improved.. I have no further concerns. I fully support publication of this manuscript.

Reviewer #2 (Remarks to the Author):

The revised manuscript by Fiore¹, Alhalaseh et al., is very impressive and reports a series of important findings that will be of interest to a wide neuroscience readership. For the authors' assurance, I was always deeply impressed by the live imaging analysis, particularly given the context of doing so in live animals, and of the very interesting finding that OPC Ca²⁺ activity is driven by locomotion. Although there remains no cell type specific demonstration that the activity-driven drive in vivo (and this was really the key point before) is mediated by specific adrenergic receptors, the authors have provided a range of complementary analyses entirely consistent with this model, and as such, I think that the manuscript is suitable for publication. My own view is that arguably the most exciting aspect of the manuscript, which was always very robust is the finding that locomotor drive regulates OPC Ca²⁺ activity and fate. I do wonder if that aspect should be infused into both the title and the abstract, certainly the latter, in which this is not noted at all. This would, I suspect, draw further wider attention to the study.

Reviewer #3 (Remarks to the Author):

The authors have addressed my previous concerns. The changes to Supplementary Figure 10 and other graphs, and the provision of additional statistical information are appreciated.

I suggest the authors now cite Lu et al. *Nature Neuroscience*, 2023 as that study includes the conditional deletion of the receptors.

Some English language editing will be required by the journal's editorial team.

We are pleased to learn that our revised manuscript addressed all the reviewers' concerns. We thank them for taking the time to evaluate our work critically and help to extend our findings further.

Reviewer #1:

Remarks to the Author:

The revision of the transferred manuscript is exceptional - the manuscript is significantly improved. I have no further concerns. I fully support publication of this manuscript.

We thank the reviewer for the kind remarks and for supporting the publication of our manuscript.

Reviewer #2

Remarks to the Author:

The revised manuscript by Fiore¹, Alhalaseh et al., is very impressive and reports a series of important findings that will be of interest to a wide neuroscience readership. For the authors' assurance, I was always deeply impressed by the live imaging analysis, particularly given the context of doing so in live animals, and of the very interesting finding that OPC Ca²⁺ activity is driven by locomotion. Although there remains no cell type specific demonstration that the activity-driven drive in vivo (and this was really the key point before) is mediated by specific adrenergic receptors, the authors have provided a range of complementary analyses entirely consistent with this model, and as such, I think that the manuscript is suitable for publication. My own view is that arguably the most exciting aspect of the manuscript, which was always very robust is the finding that locomotor drive regulates OPC Ca²⁺ activity and fate. I do wonder if that aspect should be infused into both the title and the abstract, certainly the latter, in which this is not noted at all. This would, I suspect, draw further wider attention to the study.

We thank the reviewer for the supportive and exciting comments on our work. We are pleased to see that we could address this reviewer's concerns. As recommended by the reviewer, we rewrote the abstract and now highlighted our findings on the locomotion-driven increase in the OPC Ca²⁺ transients.

Reviewer #3

Remarks to the Author:

The authors have addressed my previous concerns. The changes to Supplementary Figure 10 and other graphs, and the provision of additional statistical information are appreciated.

We are glad to learn that all the concerns were addressed, and once again, we thank the reviewer for being excited and supportive of our work.

I suggest the authors now cite Lu et al. Nature Neuroscience, 2023 as that study includes the conditional deletion of the receptors.

As suggested by the reviewer, we have now included this citation in the manuscript.

Some English language editing will be required by the journal's editorial team.

We have further edited the manuscript for the English language.